# Mitigating Catastrophic Forgetting in Large Language Models with Forgetting-aware Pruning

## Abstract

Recent advancements in large language models (LLMs) have demonstrated remarkable capabilities across a wide range of tasks. These models are typically pre-trained on extensive corpora and subsequently fine-tuned on task-specific datasets. However, during the fine-tuning process, LLMs often suffer from catastrophic forgetting, wherein previously acquired general knowledge is lost. Traditional approaches to mitigating forgetting often rely on data replay, which may not be viable when the original training data is inaccessible. Additionally, methods that alter the training process or the model architecture can increase complexity and detract from the accuracy of downstream tasks, thus limiting their generalizability. In this paper, we propose Forgetting-Aware Pruning Metric (FAPM), a novel pruning-based approach to balance forgetting and downstream task performance. Our investigation reveals that the degree to which task vectors (i.e., the subtraction of pre-trained weights from the weights fine-tuned on downstream tasks) overlap with pre-trained model parameters is a critical factor for forgetting. Motivated by this insight, FAPM employs the ratio of the task vector to pre-trained model parameters as a metric to quantify forgetting, integrating this measure into the pruning criteria. Importantly, FAPM does not necessitate modifications to the training process or model architecture, nor does it require any auxiliary data. We conducted extensive experiments across six datasets encompassing natural language inference, question answering, reading comprehension, and cloze tests. The results demonstrate that FAPM limits forgetting to just 1% while maintaining 99% accuracy on downstream tasks, rendering FAPM highly competitive relative to the state-of-the-art methods that involve modifications to the training process.

## 1 Introduction

Large language models have demonstrated impressive general capabilities in handling various tasks (Bubeck et al., 2023; Rafailov et al., 2024). Nevertheless, practical deployment frequently uncovers the necessity for augmenting domain-specific competencies (Touvron et al., 2023; Scialom et al., 2022). To this end, task-oriented datasets are harnessed to fine-tune these models, thereby enhancing their efficacy in targeted downstream tasks (Zhou et al., 2023; Yang et al., 2024b). Many studies have found that while LLMs acquire specialized knowledge during instruction fine-tuning, they tend to forget their general capabilities, especially in full fine-tuning, which is also known as Catastrophic Forgetting (CF) (Luo et al., 2023; Kong et al., 2023; Wu et al.). Consequently, devising methodologies to alleviate CF during the instruction fine-tuning phase has become a critical research direction for LLMs.

Existing methods to mitigate CF can be divided into four categories, as shown in Figure 1: 1) Replay-based methods incorporate a portion of the pre-training data into the fine-tuning data for training (Scialom et al., 2022; Huang et al., 2024). 2) Regularization-based methods introduce additional penalty terms in the loss function, encouraging the fine-tuned model to remain close to the pre-trained model (Lin et al., 2023; Panigrahi et al., 2023). 3) Weight-based methods introduce parameter weight coefficients to regulate their updates (Ke et al., 2023; Zhang et al., 2024). 4) Architecture-based methods design additional modules outside of the original model (Wang et al., 2023; Hu et al., 2021). Although these methods can alleviate the forgetting problem to a certain

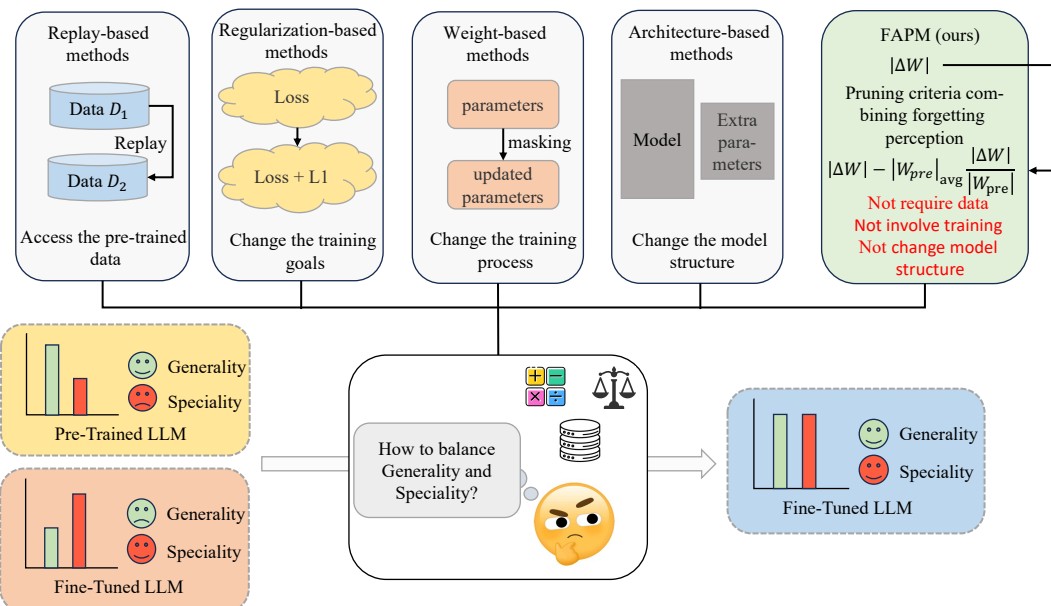

Figure 1: The diagram illustrates the issue of CF and the desired objectives. It also includes four existing methods to tackle CF, as well as our proposed FAPM.

extent, they still have the following limitations: 1) The methods that assume a certain amount of pre-training data can be obtained are unrealistic in practical applications because many open-source LLMs, e.g., Llama series, have not released their pre-training data. 2) Even if pre-training data could be obtained, incorporating it into the fine-tuning process would significantly increase training costs. 3) Methods that alter the training process or model architecture not only make the training process more difficult to control but also degrade the accuracy of downstream tasks(Ke et al., 2023; Zhang et al., 2024). The limitations of these methods lead us to think about the following question:

*Can we solve the problem of catastrophic forgetting **without changing training process**, **without any additional data**, and **without altering model structure**?*

Recent research has highlighted two key findings: 1) There are a significant number of redundant parameters in large language models (Yadav et al., 2024). 2) The task vector specifies a direction in the weight space of the pre-trained model and moving towards its direction can improve task performance (Ilharco et al., 2022). These findings suggest that we can prune portions of the task vector's parameters and set them to zero. By doing so, the corresponding positions of the pre-trained model's parameters are exposed, potentially preserving the accuracy of downstream tasks while mitigating catastrophic forgetting to some extent. To this end, we first try to apply existing pruning methods to prune the task vector to alleviate catastrophic forgetting. Unfortunately, we find it challenging to strike an optimal balance between maintaining downstream task accuracy and mitigating catastrophic forgetting using existing pruning techniques alone (Han et al., 2015; Sun et al., 2023). Specifically, pruning the task vector with a low sparsity ratio fails to effectively mitigate catastrophic forgetting, whereas pruning with a high sparsity ratio results in poor downstream task accuracy. We find that there are two main reasons for this problem: 1) The existing pruning criteria only ensure the balance between downstream task accuracy and sparsity, while not considering catastrophic forgetting. 2) The extent to which the values of task vectors overlap with pre-trained model parameters is a critical factor contributing to catastrophic forgetting.

In this paper, we propose a new pruning method called Forgetting-Aware Pruning Metric (FAPM). FAPM not only applies magnitude as the pruning criterion for task vectors but also uses the ratio of task vectors to pre-trained model parameters as the criterion for mitigating CF. By adopting FAPM, we aim to identify, during the pruning process, those parameters in the task vector whose values are large (crucial for maintaining the accuracy of downstream tasks) and we concurrently intend to penalize those parameters where the ratio of their magnitude in the task vector to that of the corresponding parameter in the pre-trained model is notably high (more likely to induce CF). This

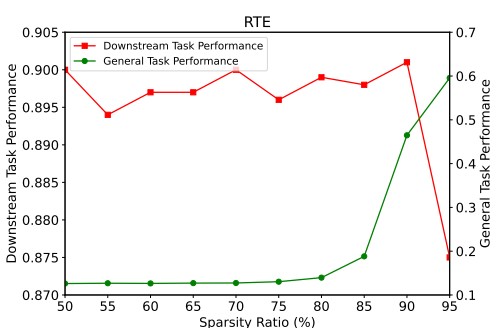 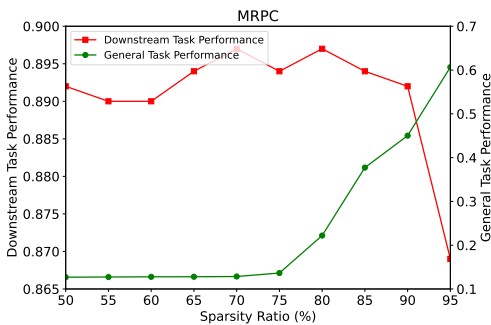

(a) The original accuracy on RTE is 0.890 and the original average accuracy on four general tasks is 0.6204.

(b) The original accuracy on MRPC is 0.887 and the original average accuracy on four general tasks is 0.6204.

Figure 2: The relationship between the magnitude pruning sparsity ratio, general capability, and downstream task performance of Llama3-8B on (a) RTE and (b) MRPC, respectively. When sparsity is below 90%, downstream task performance remains relatively stable, but CF is notably serious (general task performance is poor). When sparsity exceeds 90%, increasing sparsity alleviates CF effectively, but significantly reduces downstream task performance. Consequently, achieving an optimal balance between downstream task performance and CF via magnitude pruning is hard.

balanced approach aspires to surgically retain the most valuable parameters for task performance while excising those that pose the greatest risk to the model's generality. Extensive experiments on different LLMs and various datasets show that FAPM can maintain a downstream task accuracy of up to 99% while the degree of CF is only 1%. Compared to structure-based strategies, such as LoRA, FAPM not only achieves superiority in precision but also maintains the same level of forgetting rate. Compared to other methods that adjust training strategies, it also demonstrates strong competitiveness.

## 2 BACKGROUND AND MOTIVATION

**Problem Setting.** Given downstream data $D$ and a pre-trained model like Llama3, we fine-tune the model using $D$. Let the pre-trained model parameters be $W_{pre}$ and the fine-tuned model parameters be $W_{ft}$. In this paper, we perform a series of operations on the task vector. Following previous work (Ilharco et al., 2022), for a task, the task vector $\Delta W \in \mathbb{R}_d$ can be defined as $W_{ft} - W_{pre}$. This operation allows us to focus on the changes that occur during the fine-tuning stage.

**Pruning on the task vector.** We first try to apply existing pruning methods to prune the task vector, which prunes parameters in the task vector according to their magnitude (Han et al., 2015). The red lines in Figure 2(a) and Figure 2(b) display how downstream performance changes with sparsity ratio on RTE and MRPC datasets. For each sparsity ratio, the model task vectors are "trimmed" to retain only the top-k% highest-magnitude values, with the remaining values reset to zero. The green lines in Figure 2 illustrate the impact of the sparsity ratio of the task vector on general task performance, focusing particularly on the extent of CF, where higher accuracy indicates lesser forgetting.

From Figure 2, we find that numerous values in a given task vector are redundant, and their removal does not compromise task accuracy. Remarkably, the downstream task accuracy remains stable even when the sparsity ratio reaches 90%. This suggests that when the pruned $\Delta W$ and $W_{pre}$ are combined as the final fine-tuning parameters, 90% parameters in $W_{pre}$ are exposed. Conversely, when sparsity exceeds 90%, increasing sparsity can further alleviate CF effectively; however, this coincides with a marked deterioration in downstream task performance. This paradox highlights the inherent difficulty in striking an ideal balance between maintaining downstream task performance and mitigating CF solely through magnitude pruning on $\Delta W$. This raises a pertinent question:

*What additional factors, beyond $\Delta W$ itself, could also affect the balance between maintaining downstream task accuracy and mitigating CF?*

# 3 FAPM: FORGETTING-AWARE PRUNING METRIC

## 3.1 EXPLORATION AND ANALYSIS

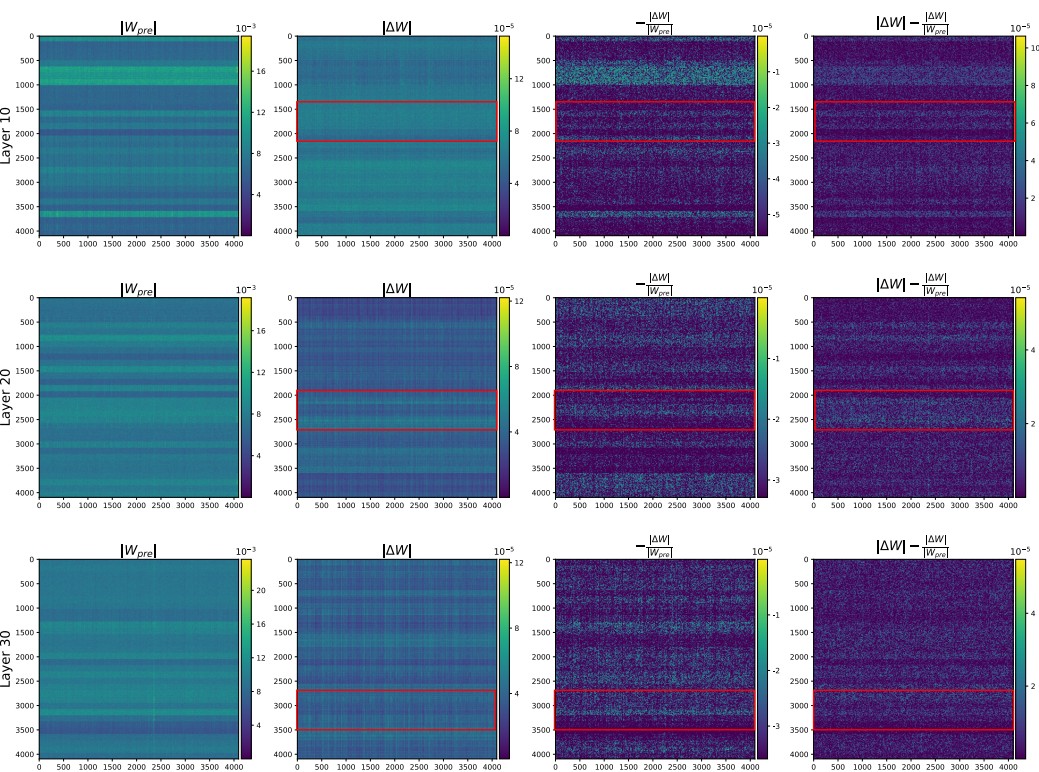

Figure 3: Visualization of the weight matrices in different layers of Llama3-8B fine-tuned on RTE dataset. From left to right, they represent the magnitude of the pre-trained model weights, the absolute change magnitude of model weights, the relative change magnitude of model weights, and a combination of the absolute and relative change magnitude. The absolute and relative changes patterns show clear differences, such as the channels marked by the red boxes.

Considering $W_{ft} = W_{pre} + \Delta W$, we hypothesize that the relative magnitude between $\Delta W$ and $W_{pre}$ is also a crucial factor influencing the balance between downstream task accuracy and forgetting. Let's consider a simple scenario: if the change in parameters during the fine-tuning process is zero, then the fine-tuned model equals the pre-trained model, making this model optimal for addressing the forgetting problem, i.e., there is no forgetting. Conversely, if the change in the pre-trained model parameters during fine-tuning is substantial, it indicates a significant modification of the pre-trained model parameters. The greater the difference between the new model and the no-forgetting model, the more likely it is to result in forgetting.

Intuitively, at a certain parameter position, if the ratio of the magnitudes $\frac{|\Delta W|}{|W_{pre}|}$ is larger, it suggests that the fine-tuning process has a greater impact on the parameters at that position, and thus, it is more likely to cause forgetting. We refer to this influencing factor as the "relative change magnitude" and refer to the criterion that prunes solely based on the magnitude of $\Delta W$ as "absolute change magnitude" criterion. Compared to absolute change magnitude criterion, the relative change magnitude models the relative relationship between $\Delta W$ and $W_{pre}$. Since the pre-trained model $W_{pre}$ is crucial for mitigating forgetting, this criterion better reflects which parameters in $\Delta W$ are more critical for mitigating forgetting.

In Figure 3, we illustrate the differences in attention to various positions within the model weight matrices across different layers, guided by the absolute change magnitude criterion and the relative change magnitude criterion. Brighter regions in the figure represent parameters with higher scores

under a specific criterion, while darker regions denote parameters with lower scores. Under the absolute change magnitude criterion, the highlighted areas indicate parameters crucial for downstream task accuracy. In contrast, under the relative change magnitude criterion, the highlighted regions indicate parameters important for mitigating forgetting.

By comparing the images in the middle two columns, we observe a significant divergence in scoring patterns between the absolute change magnitude criterion and the relative change magnitude criterion. The highlighted areas under the absolute change magnitude criterion do not entirely correspond to those under the relative change magnitude criterion. This discrepancy indicates that parameters retained under the absolute change magnitude criterion may not be effective in mitigating catastrophic forgetting. This also explains why it is difficult to balance downstream task accuracy and forgetting when using $|\Delta W|$ as the pruning criterion alone. To achieve a more favorable balance between downstream task accuracy and CF, we propose a hybrid pruning criterion that incorporates both the absolute change magnitude and relative change magnitude. This combined criterion fuses the strengths of both individual criteria and exhibits a distinct pattern that differs from using either criterion in isolation as shown in the rightmost part of Figure 3.

## 3.2 Pruning Metric

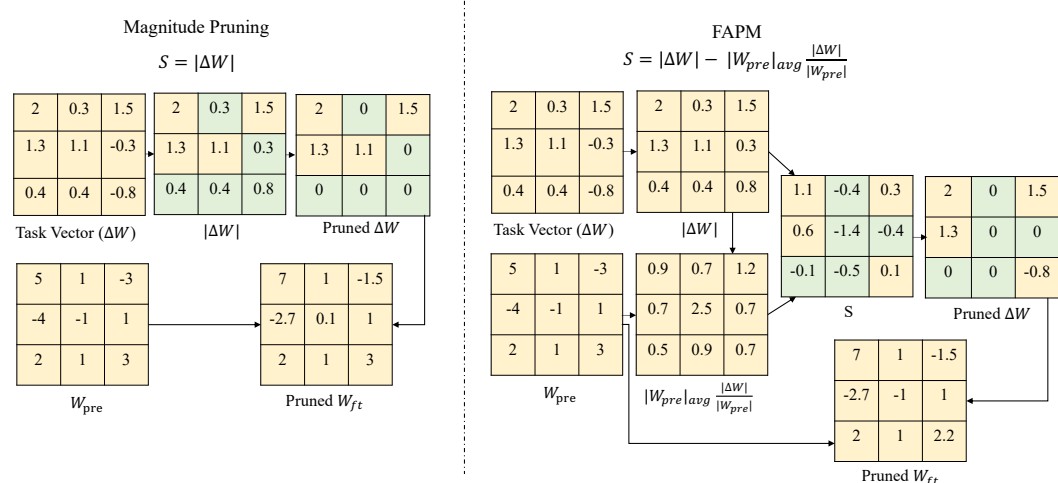

Figure 4: Illustration of FAPM, compared with the magnitude pruning. If $|\Delta W^i|$ is large (indicating it will be retained by magnitude pruning) and $\frac{|\Delta W^i|}{|W_{pre}^i|}$ is also large (suggesting it may contribute CF), our FAPM will penalize and possibly prune this parameter, e.g., the value 1.1 in the middle of $\Delta W$. By doing so, most large-magnitude parameters in $\Delta W$ are retained, while only a small subset are replaced by parameters with smaller $\frac{|\Delta W^i|}{|W_{pre}^i|}$ values.

Consider a linear layer in $\Delta W$ with weights $\Delta W^i$ of shape $(C_{in}, C_{out})$, corresponding to the linear layer representation $W_{pre}^i$ in $W_{pre}$. We propose to evaluate each weight matrix's importance by subtracting the relative change magnitude criterion from the absolute change magnitude criterion. Specifically, the pruning criterion for $\Delta W^i$ is defined as follows:

$$S_i = |\Delta W^i| - |W_{pre}^i|_{avg} \frac{|\Delta W^i|}{|W_{pre}^i|} \tag{1}$$

where $|\cdot|$ denotes the absolute value operation, and $avg$ represents the averaging operation on the parameter matrix. $i$ denotes one of the matrices in the $\Delta W$ parameter matrix. We included $|W_{pre}^i|_{avg}$ in the formula due to our observations during practical operations. We found that the numerical values of $\frac{|\Delta W^i|}{|W_{pre}^i|}$ and $|\Delta W^i|$ do not fall within the same range. For instance, the order of magnitude of $\frac{|\Delta W^i|}{|W_{pre}^i|}$ is approximately $1 \times 10^{-2}$, whereas that of $|\Delta W^i|$ is around $1 \times 10^{-4}$. This will lead one criterion to predominate over the other, weakening the impact of the other. Therefore, to balance the

numerical values of $\frac{|\Delta W^i|}{|W^i_{pre}|}$ and $|\Delta W^i|$, we have introduced $|W^i_{pre}|_{avg}$. The specific pruning process of FAPM can be seen in Figure 4.

Our FAPM has several intriguing properties. Firstly, when the value of a parameter in $|\Delta W^i|$ is large (indicating that the parameter would typically be retained according to traditional magnitude pruning criteria) and simultaneously, $\frac{|\Delta W^i|}{|W^i_{pre}|}$ is also large (suggesting that this parameter may contribute to catastrophic forgetting), the FAPM pruning strategy will penalize and potentially prune this parameter. Under the FAPM criteria, to ensure downstream task accuracy, most parameters with large magnitudes will still be retained, while only a small subset will be replaced by parameters with smaller $\frac{|\Delta W^i|}{|W^i_{pre}|}$ values. Secondly, the computation of FAPM is both simple and efficient. We only need to obtain the fine-tuned and pre-trained model parameters, eliminating the need for additional data. The computational overhead associated with this method is minimal, enhancing the generalizability of FAPM. We provide the pseudocode implementation of FAPM in Appendix A.

## 4 EXPERIMENT SETUP

**Models and Evaluation:** We evaluate FAPM on two widely adopt LLMs: Llama3-8B (Dubey et al., 2024) and Qwen2-7B (Yang et al., 2024a). Following previous studies (Yadav et al., 2024; Wu et al., 2024; Han et al., 2024), we evaluate FAPM's specialized performance across four key tasks: natural language inference, question answering, cloze tests, and reading comprehension. We utilize the MRPC (3.67k training samples) (Wang et al., 2019) and RTE (2.49k training samples) (Wang et al., 2019) datasets for natural language inference, with accuracy as the evaluation metric. For question answering, we employ the WikiQA (20.4k training samples) (Yang et al., 2015) and QASC (8.13k training samples) datasets (Khot et al., 2020), using ROUGE-L as the evaluation metric. We use the Winogrande dataset (10.2k training samples) (Sakaguchi et al., 2021) for cloze tests, measuring performance with accuracy. Lastly, we utilize the SQuAD dataset (87.6k training samples) (Rajpurkar et al., 2016) for reading comprehension, with the F1-score as the evaluation metric.

To evaluate the generality of LLMs, we integrate insights from previous studies (Dubey et al., 2024; Yang et al., 2024a) and focus on four key aspects. We use MMLU (Hendrycks et al., 2021) to assess the inherent world knowledge stored in the LLM, C-Eval (Huang et al., 2023) to evaluate the model's understanding of general knowledge in Chinese, GSM8K (Cobbe et al., 2021) to evaluate mathematical reasoning, and HumanEval (Chen et al., 2021) to assess the code generation capabilities.

**Compared Methods** We compared FAPM with the full-parameter SFT (Full SFT) and four CF baselines, which are described in detail in Appendix B. These baselines are carefully categorized into three groups: 1) Regularization-based: These methods introduce additional terms in the loss function to constrain parameter changes. The selected comparison baselines are L1 regularization (Kirkpatrick et al., 2017). 2) Weight-based: These methods design a coefficient for each weight to control its update during training. The selected baselines include V-SoftMask (Ke et al., 2023) and CoFiTune (Zhang et al., 2024). 3) Architecture-based: These methods introduce additional parameters to ensure the pre-trained model's parameters remain frozen during training. The selected baseline is LoRA (Hu et al., 2021).

**Experimental Setting:** During training, we set the learning rate to 5e-6 and the batch size to 2. Each dataset was trained for 3 epochs. The AdamW optimizer was used for fine-tuning. We employed LLaMA-Factory (Zheng et al., 2024) as the training platform and vLLM (Kwon et al., 2023) for inference. When implementing the FAPM algorithm, we applied a 90% sparsity rate across all models and datasets.

## 5 RESULTS

In this section, we aim to investigate the effectiveness of the FAPM method in maintaining generalization capabilities while learning downstream tasks. We fine-tune the Llama3-8B (Dubey et al., 2024) and Qwen2-7B (Yang et al., 2024a) models on MRPC (Wang et al., 2019), RTE (Wang et al., 2019), WikiQA (Yang et al., 2015), Winogrande (Sakaguchi et al., 2021), QASC (Khot et al., 2020), and SQuAD datasets (Rajpurkar et al., 2016). The performance of FAPM is compared against sev-

| Tasks | Methods | C-Eval | GSM8K | MMLU | HumanEval | Avg. | Performance |
|---|---|---|---|---|---|---|---|
| | Pre-trained model | 0.4386 | 0.7922 | 0.6594 | 0.5914 | 0.6204 | 0.819 |
| | Full SFT | 0.2311 | 0.075 | 0.2554 | 0.0 | 0.1403 | 0.890 |
| RTE | L1-reg (Kirkpatrick et al., 2017) | 0.3735 | 0.7353 | 0.6012 | 0.5367 | 0.5616 | 0.843 |
| | V-SoftMask (Ke et al., 2023) | 0.4144 | 0.7811 | 0.5702 | 0.4919 | 0.5644 | 0.886 |
| | CoFiTune (Zhang et al., 2024) | 0.4542 | 0.7869 | 0.6492 | 0.5815 | 0.6180 | 0.882 |
| | LoRA (Hu et al., 2021) | 0.4435 | 0.7892 | 0.6574 | 0.5915 | 0.6204 | 0.866 |
| | FAPM (Ours) | 0.4623 | 0.7915 | 0.6454 | 0.5975 | **0.6242** | **0.897** |
| | Pre-trained model | 0.4386 | 0.7922 | 0.6594 | 0.5914 | 0.6204 | 0.913 |
| | Full SFT | 0.2547 | 0.0 | 0.2422 | 0.0 | 0.1242 | 0.966 |
| WikiQA | L1-reg (Kirkpatrick et al., 2017) | 0.4271 | 0.7591 | 0.5780 | 0.5549 | 0.5797 | 0.945 |
| | V-SoftMask (Ke et al., 2023) | 0.2944 | 0.7282 | 0.5677 | 0.2910 | 0.4703 | 0.963 |
| | CoFiTune (Zhang et al., 2024) | 0.4164 | 0.7702 | 0.6309 | 0.5666 | 0.5960 | 0.960 |
| | LoRA (Hu et al., 2021) | 0.4423 | 0.8013 | 0.6429 | 0.5919 | 0.6196 | 0.955 |
| | FAPM (Ours) | 0.4749 | 0.7975 | 0.6563 | 0.5853 | **0.6285** | **0.964** |
| | Pre-trained model | 0.4386 | 0.7922 | 0.6594 | 0.5914 | 0.6204 | 0.519 |
| | Full SFT | 0.2792 | 0.0606 | 0.3438 | 0.0 | 0.1709 | 0.820 |
| Winogrande | L1-reg (Kirkpatrick et al., 2017) | 0.4234 | 0.7572 | 0.6245 | 0.5667 | 0.5904 | 0.737 |
| | V-SoftMask (Ke et al., 2023) | 0.4089 | 0.7017 | 0.5528 | 0.5003 | 0.5409 | **0.828** |
| | CoFiTune (Zhang et al., 2024) | 0.4719 | 0.7817 | 0.6410 | 0.5743 | 0.6172 | 0.813 |
| | LoRA (Hu et al., 2021) | 0.4622 | 0.7922 | 0.6429 | 0.5975 | **0.6237** | 0.810 |
| | FAPM (Ours) | 0.4829 | 0.7680 | 0.6472 | 0.5731 | 0.6178 | 0.824 |
| | Pre-trained model | 0.4386 | 0.7922 | 0.6594 | 0.5914 | 0.6204 | 0.371 |
| | Full SFT | 0.2806 | 0.0212 | 0.3206 | 0.0 | 0.1556 | 0.646 |
| SQuAD | L1-reg (Kirkpatrick et al., 2017) | 0.3990 | 0.6605 | 0.5800 | 0.5113 | 0.5377 | 0.565 |
| | V-SoftMask (Ke et al., 2023) | 0.3757 | 0.0786 | 0.4755 | 0.5013 | 0.3578 | 0.635 |
| | CoFiTune (Zhang et al., 2024) | 0.4619 | 0.7596 | 0.6356 | 0.5766 | **0.6084** | 0.633 |
| | LoRA (Hu et al., 2021) | 0.4795 | 0.7255 | 0.5914 | 0.5853 | 0.5954 | **0.648** |
| | FAPM (Ours) | 0.4738 | 0.7310 | 0.6455 | 0.5748 | 0.6063 | 0.637 |

Table 1: The comparison results of FAPM and different baselines on various datasets using the Llama3-8B model. "Avg." represents the average results across the C-Eval, GSM8K, MMLU, and HumanEval datasets. "Performance" indicates the accuracy on the respective downstream task datasets.

eral baseline methods. The evaluation focuses on performance changes in downstream tasks and generalization ability metrics, using the performance of Full SFT and the pre-trained models as reference points.

## 5.1 COMPARATIVE ANALYSIS OF FAPM AGAINST VARIOUS BASELINES

In Tables 1 and 2, we present the comparative results of FAPM and various baselines on different datasets, using the Llama3-8B and Qwen2-7B models. Due to space constraints, in the main text, we have chosen one dataset from each of the four downstream tasks for presentation. For natural language inference, we selected the RTE dataset, and for question answering, we chose the WikiQA dataset. The experimental results for MRPC and QASC can be found in Appendix C.

Table 1 shows that Full SFT exhibits significant forgetting on Llama3-8B, with average accuracy on four general datasets maintaining only around 0.15. This indicates that the fine-tuned model loses almost all generalization capability, demonstrating that full fine-tuning severely exacerbates catastrophic forgetting. On the Llama3-8B model, FAPM achieves an average performance of 0.8445 on six downstream datasets, compared to Full SFT's 0.8454, indicating that FAPM has minimal impact on downstream task performance. Furthermore, FAPM's average performance on the four general tasks is 0.6196, a decrease of only 0.08% compared to the Pre-trained model, demonstrating that FAPM significantly alleviates forgetting. Similar trends are observed with the Qwen2-7B model as shown in Table 2. These results indicate that our proposed FAPM method effectively maintains downstream task performance while alleviating catastrophic forgetting.

Compared to L1-regularization, FAPM demonstrates a stronger ability to preserve downstream task accuracy and better addresses catastrophic forgetting. Specifically, for the Llama3-8B model, L1-regularization results in an average performance drop of 5.99% across six downstream datasets. While both LoRA and FAPM similarly mitigate catastrophic forgetting, LoRA slightly compromises

| Tasks | Methods | C-Eval | GSM8K | MMLU | HumanEval | Avg. | Performance |
|---|---|---|---|---|---|---|---|
| | Pre-trained model | 0.7478 | 0.8180 | 0.6884 | 0.7682 | 0.7556 | 0.574 |
| | Full SFT | 0.2602 | 0.075 | 0.2423 | 0.0 | 0.1443 | 0.890 |
| RTE | L1-reg (Kirkpatrick et al., 2017) | 0.7108 | 0.7463 | 0.6143 | 0.7118 | 0.6958 | 0.847 |
| | V-SoftMask (Ke et al., 2023) | 0.7317 | 0.7371 | 0.6448 | 0.7111 | 0.7062 | 0.896 |
| | CoFiTune (Zhang et al., 2024) | 0.7591 | 0.8125 | 0.6808 | 0.7560 | **0.7521** | 0.886 |
| | LoRA (Hu et al., 2021) | 0.7456 | 0.8133 | 0.6897 | 0.7500 | 0.7496 | 0.877 |
| | FAPM (Ours) | 0.7568 | 0.8104 | 0.6857 | 0.7500 | 0.7507 | **0.903** |
| | Pre-trained model | 0.7478 | 0.8180 | 0.6884 | 0.7682 | 0.7556 | 0.896 |
| | Full SFT | 0.2510 | 0.076 | 0.2434 | 0.0 | 0.1426 | 0.965 |
| WikiQA | L1-reg (Kirkpatrick et al., 2017) | 0.6818 | 0.7582 | 0.6186 | 0.7091 | 0.6919 | 0.955 |
| | V-SoftMask (Ke et al., 2023) | 0.6862 | 0.6585 | 0.5331 | 0.6759 | 0.6384 | **0.965** |
| | CoFiTune (Zhang et al., 2024) | 0.7527 | 0.7755 | 0.6358 | 0.7195 | 0.7208 | 0.961 |
| | LoRA (Hu et al., 2021) | 0.7519 | 0.8119 | 0.6873 | 0.7621 | **0.7533** | 0.960 |
| | FAPM (Ours) | 0.7555 | 0.8036 | 0.6902 | 0.7621 | 0.7529 | 0.962 |
| | Pre-trained model | 0.7478 | 0.8180 | 0.6884 | 0.7682 | 0.7556 | 0.558 |
| | Full SFT | 0.4090 | 0.0303 | 0.2996 | 0.0609 | 0.1999 | 0.790 |
| Winogrande | L1-reg (Kirkpatrick et al., 2017) | 0.7283 | 0.7609 | 0.6401 | 0.7277 | 0.7143 | 0.703 |
| | V-SoftMask (Ke et al., 2023) | 0.7321 | 0.7098 | 0.6241 | 0.6861 | 0.6880 | **0.791** |
| | CoFiTune (Zhang et al., 2024) | 0.7550 | 0.7990 | 0.6820 | 0.7500 | 0.7465 | 0.771 |
| | LoRA (Hu et al., 2021) | 0.7530 | 0.8118 | 0.6861 | 0.7500 | **0.7502** | 0.782 |
| | FAPM (Ours) | 0.7618 | 0.8068 | 0.6845 | 0.7395 | 0.7482 | 0.785 |
| | Pre-trained model | 0.7478 | 0.8180 | 0.6884 | 0.7682 | 0.7556 | 0.451 |
| | Full SFT | 0.3531 | 0.0212 | 0.3183 | 0.0 | 0.1731 | 0.624 |
| SQuAD | L1-reg (Kirkpatrick et al., 2017) | 0.6481 | 0.6614 | 0.5883 | 0.6681 | 0.6414 | 0.561 |
| | V-SoftMask (Ke et al., 2023) | 0.6369 | 0.5881 | 0.5933 | 0.6451 | 0.6159 | **0.624** |
| | CoFiTune (Zhang et al., 2024) | 0.7451 | 0.7626 | 0.6584 | 0.7621 | 0.7321 | 0.619 |
| | LoRA (Hu et al., 2021) | 0.7253 | 0.7665 | 0.6537 | 0.7482 | 0.7234 | 0.620 |
| | FAPM (Ours) | 0.7410 | 0.8006 | 0.6752 | 0.7500 | **0.7417** | 0.615 |

Table 2: The results of FAPM and different baselines on various datasets on Qwen2-7B.

downstream task accuracy, particularly on the MRPC and RTE datasets. V-SoftMask excels in preserving downstream task accuracy but performs poorly in addressing catastrophic forgetting, with an average performance drop of 10.92% on four general tasks. Compared to the CoFiTune method, FAPM also demonstrates comparable performance. Overall, FAPM shows strong competitiveness in both maintaining downstream task accuracy and mitigating catastrophic forgetting when compared to existing regularization-based, weight-based, and architecture-based methods.

## 5.2 COMPARATIVE ANALYSIS OF DIFFERENT PRUNING CRITERIA

One question that needs to be analyzed is why FAPM is improved based on Magnitude Pruning instead of the SOTA LLM pruning method. To elucidate this choice, this section examines the application of a straightforward and efficient pruning method, Wanda (Sun et al., 2023), to mitigate the issue of catastrophic forgetting. Wanda addresses pruning by removing weights with the smallest magnitudes, as determined by the product of the weight magnitudes and the norms of the corresponding input activations, thereby preventing the need for retraining or weight updates, which is formulated as $S_{ij} = |W_{ij}| \cdot \|X_j\|_2$. We prune $\Delta W$ according to this criterion in our experiments.

Table 3 and Table 4 present the comparative results of FAPM and different pruning criteria methods, with all results using a 90% sparsity ratio. These tables reveal that while Wanda can somewhat mitigate catastrophic forgetting, it significantly impairs performance on downstream tasks. For example, in the Llama3-8B model, Wanda results in an average performance decline of 3.6% across six downstream datasets when compared to Full SFT, whereas Magnitude Pruning exhibits negligible impact on downstream task accuracy. Given the necessity to preserve downstream task accuracy, we have opted to use Magnitude Pruning as our foundational pruning criterion. Furthermore, Wanda requires a small amount of calibration data while Magnitude Pruning does not necessitate any auxiliary data. This further reinforces our decision to select Magnitude Pruning as the basis for our methodology. More experimental results on MRPC and QASC can be found in Appendix C.

| Tasks | Methods | C-Eval | GSM8K | MMLU | HumanEval | Avg. | Performance |
|---|---|---|---|---|---|---|---|
| RTE | Magnitude (Han et al., 2015) | 0.3063 | 0.6631 | 0.4052 | 0.4843 | 0.4647 | **0.901** |
| | Wanda (Sun et al., 2023) | 0.4675 | 0.7827 | 0.6465 | 0.5732 | 0.6174 | 0.878 |
| | FAPM (Ours) | 0.4623 | 0.7915 | 0.6454 | 0.5975 | **0.6242** | 0.897 |
| WikiQA | Magnitude (Han et al., 2015) | 0.2606 | 0.0 | 0.2553 | 0.0 | 0.1289 | 0.964 |
| | Wanda (Sun et al., 2023) | 0.4760 | 0.7804 | 0.6432 | 0.5834 | 0.6207 | 0.961 |
| | FAPM (Ours) | 0.4749 | 0.7975 | 0.6563 | 0.5853 | **0.6285** | **0.964** |
| Winogrande | Magnitude (Han et al., 2015) | 0.4957 | 0.6148 | 0.6236 | 0.5731 | 0.5768 | **0.828** |
| | Wanda (Sun et al., 2023) | 0.4748 | 0.7762 | 0.6508 | 0.5919 | 0.6234 | 0.750 |
| | FAPM (Ours) | 0.4829 | 0.7680 | 0.6472 | 0.5731 | 0.6178 | 0.824 |
| SQuAD | Magnitude (Han et al., 2015) | 0.4504 | 0.1 | 0.5816 | 0.1951 | 0.3318 | **0.641** |
| | Wanda (Sun et al., 2023) | 0.4648 | 0.6686 | 0.6284 | 0.3536 | 0.5288 | 0.611 |
| | FAPM (Ours) | 0.4738 | 0.7310 | 0.6455 | 0.5748 | **0.6063** | 0.637 |

Table 3: The results of FAPM and different pruning methods on various datasets on Llama3-8B.

| Tasks | Methods | C-Eval | GSM8K | MMLU | HumanEval | Avg. | Performance |
|---|---|---|---|---|---|---|---|
| RTE | Magnitude (Han et al., 2015) | 0.7144 | 0.7346 | 0.5190 | 0.6943 | 0.6655 | 0.895 |
| | Wanda (Sun et al., 2023) | 0.7442 | 0.8025 | 0.6774 | 0.7542 | 0.7445 | 0.877 |
| | FAPM (Ours) | 0.7568 | 0.8104 | 0.6857 | 0.7500 | **0.7507** | **0.903** |
| WikiQA | Magnitude (Han et al., 2015) | 0.7162 | 0.0416 | 0.2560 | 0.0 | 0.2535 | **0.965** |
| | Wanda (Sun et al., 2023) | 0.7520 | 0.7793 | 0.6784 | 0.7134 | 0.7307 | 0.958 |
| | FAPM (Ours) | 0.7555 | 0.8036 | 0.6902 | 0.7621 | **0.7529** | 0.962 |
| Winogrande | Magnitude (Han et al., 2015) | 0.6849 | 0.5056 | 0.6133 | 0.5975 | 0.6003 | 0.742 |
| | Wanda (Sun et al., 2023) | 0.7549 | 0.7915 | 0.6806 | 0.5914 | 0.7046 | 0.731 |
| | FAPM (Ours) | 0.7618 | 0.8068 | 0.6845 | 0.7395 | **0.7482** | **0.785** |
| SQuAD | Magnitude (Han et al., 2015) | 0.7189 | 0.1501 | 0.6135 | 0.0976 | 0.3950 | 0.588 |
| | Wanda (Sun et al., 2023) | 0.7315 | 0.4291 | 0.6573 | 0.3170 | 0.5337 | 0.533 |
| | FAPM (Ours) | 0.7410 | 0.8006 | 0.6752 | 0.7500 | **0.7417** | **0.615** |

Table 4: The results of FAPM and different pruning methods on various datasets on Qwen2-7B.

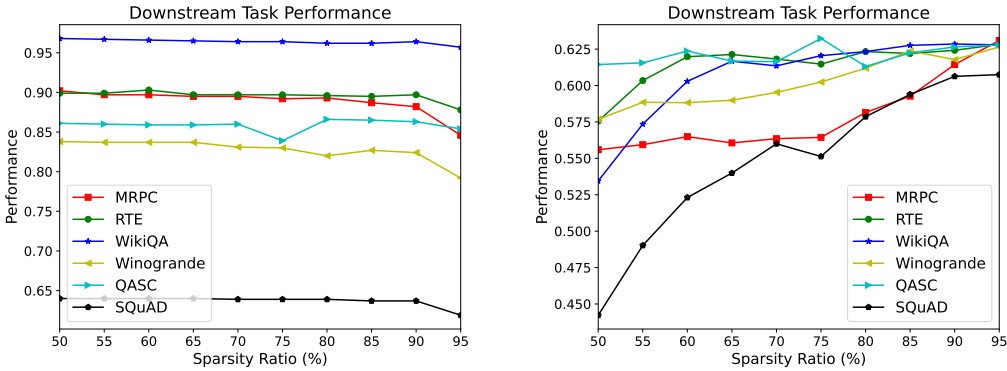

Figure 5: Performance of FAPM on downstream task accuracy and mitigation of catastrophic forgetting with different sparsity ratios on Llama3-8B.

### 5.3 Effects of Sparsity

In this section, we explore the performance of FAPM under different sparsity ratios. Figure 5 shows the impact of FAPM on downstream task accuracy and catastrophic forgetting at different sparsity ratios on Llama3-8B. As observed in Figure 2, using $|\Delta W|$ as the pruning criterion results in severe catastrophic forgetting at an 85% sparsity ratio. However, with the application of FAPM, catastrophic forgetting is substantially mitigated even at the 85% sparsity level. Notably, FAPM continues to alleviate catastrophic forgetting to some extent at a 55% sparsity ratio in the QASC and RTE datasets, highlighting its effectiveness in preventing catastrophic forgetting. Moreover, it was observed that downstream task accuracy significantly declines when the sparsity ratio exceeds 90%. Conversely, when the sparsity ratio is maintained below 90%, the impact on downstream task accuracy is minimal, although the incidence of catastrophic forgetting gradually increases. These observations suggest that a 90% sparsity ratio may represent an optimal balance, preserving downstream

task accuracy while minimizing catastrophic forgetting. More experimental results on Qwen2-7B are presented in Appendix C.

## 6 Related Work

**Catastrophic Forgetting in LLMs.** Fine-tuning LLMs with additional task-specific data, a common practice to enhance model specialization, often leads to catastrophic forgetting of previously acquired general capabilities (Luo et al., 2023; Kong et al., 2023; Wu et al.). Existing approaches to mitigate catastrophic forgetting can be broadly categorized into four main categories: 1) Replay-based methods (Scialom et al., 2022; Huang et al., 2024) typically integrate some pre-training data into the fine-tuning dataset for training. However, the assumption of access to a certain amount of pre-training data is often unrealistic in practice. 2) Regularization-based methods (Lin et al., 2023; Panigrahi et al., 2023) introduce additional penalty terms in the loss function, encouraging the fine-tuned model to maintain proximity to the pre-trained model. 3) Weight-based methods (Ke et al., 2023; Zhang et al., 2024) introduce parameter weight coefficients to modulate their updates, thereby ensuring controlled adjustments during the optimization process. However, both regularization-based and weight-based methods require to modify the optimization process, which makes the training process more challenging. 4) Architecture-based methods (Wang et al., 2023; Hu et al., 2021; Razdaibiedina et al., 2023) involve the design of additional modules external to the original model. These methods enhance models' specialization without altering the core architecture but their effects on improving downstream task accuracy are limited.

**LLM Pruning.** Network pruning (LeCun et al., 1989; Han et al., 2015), which shrinks network sizes by removing specific weights, is often considered a popular approach for compressing LLMs. Magnitude Pruning (Han et al., 2015) is a standard pruning technique to induce sparsity in models. It removes individual weights based on their magnitudes, where weights with magnitudes below a certain threshold are removed. Recent LLM pruning methods typically involve calculating pruning metrics according to model weights and activations by using some additional data. SparseGPT (Frantar & Alistarh, 2023) frames pruning as an extensive sparse regression problem and solves it using an approximate sparse regression solver. Wanda (Sun et al., 2023) prunes weights with the smallest magnitudes multiplied by the norm of the corresponding input activations, without the need for retraining or weight updates. DSnoT (Zhang et al., 2023) minimizes the reconstruction error between dense and sparse models through iterative weight pruning and growing. All these methods aim to increase the sparsity of the model as much as possible and reduce the model parameters while maintaining model performance. Different from this, in this paper, we intend to achieve a better balance mitigating CF and improving downstream accuracy by pruning task vectors in LLM fine-tuning.

## 7 Conclusion

In this study, we present a straightforward and efficient method to tackle the issue of catastrophic forgetting that emerges during the continuous fine-tuning of LLMs. Inspired by the magnitude-based pruning techniques employed in LLMs, we propose a new pruning criterion, known as the Forgetting-Aware Pruning Metric, which effectively addresses catastrophic forgetting while preserving the performance of the fine-tuning tasks. Our research reveals that the extent to which task vectors overlap with the pre-trained model parameters is a key factor influencing catastrophic forgetting. Based on this insight, FAPM integrates the ratio of the task vector to the pre-trained model parameters as a criterion, combining it with the magnitude-based pruning metric. Our FAPM does not require any additional auxiliary data, nor does it necessitate alterations to the training process or model structure. It operates solely during the inference phase, thereby enhancing its versatility. We hope our work serves as a baseline for future research in this area and encourages further exploration into understanding CF during the inference phase of LLMs.

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

# A    PSEUDOCODE FOR FAPM

In this section, we describe the pseudocode for FAPM. A detailed introduction to FAPM can be found in Section 3 of the main paper.

---

**Algorithm 1** FAPM Procedure

---

**Input:** pre-trained model $W_{pre}$, fine-tuned model $W_{ft}$, layer number $L$, desired sparsity $s$.
**Output:** pruned $W_{ft}^i$.
  **for** $i \in [0, L]$ **do**
      $\Delta W^i = W_{ft}^i - W_{pre}^i$.
      Calculate score vector $S^i \leftarrow |\Delta W^i| - \text{Avg}(|W_{pre}^i|) * \frac{|\Delta W^i|}{|W_{pre}^i|}$.
      Obtain pruning threshold $t^i$ according to $s$ and $S^i$.
      Obtain pruning mask matrix $M^i = \mathbf{1}[\![S^i > t^i]\!]$.
      $\Delta W^i \leftarrow \Delta W^i \odot M^i$.
      $W_{ft}^i = W_{pre}^i + \Delta W^i$.
  **end for**

---

# B    BASELINE DESCRIPTIONS

In this Section, we describe the baseline method in our setting in detail.

L1 regularization (Kirkpatrick et al., 2017) adds an L1 penalty term to the original loss function to promote sparsity in the parameter updates. The modified loss function is $L(\theta) + \lambda_1 \|\theta - \theta_{pre}\|_1$, with the regularization hyperparameter set to 1e-6.

Ke et al. (Ke et al., 2023) proposed the Vanilla Soft-masking method to address the issue of catastrophic forgetting in language models during continual fine-tuning. Specifically, this method employs a gradient-based detection technique to calculate the importance of units within both the attention and feed-forward network (FFN) modules across all transformer layers. The obtained importance weights are then used to control the backpropagation of the gradients.

Zhang et al. (Zhang et al., 2024) proposed the CoFiTune method to tackle the issue of catastrophic forgetting. CoFiTune employs a two-stage approach. At the coarse-grained level, an empirical tree-search algorithm is used to identify and update specific modules that are crucial for the fine-tuning task, while keeping other parameters frozen. At the fine-grained level, a soft-masking mechanism is employed to adjust the updates of the large model, thereby alleviating catastrophic forgetting.

Inspired by the perspective that "pre-trained models have a lower intrinsic dimension when fine-tuned on specific tasks," Hu et al. (Hu et al., 2021) proposed a fine-tuning method called LoRA. During the training process of LoRA, the pre-trained parameters are kept frozen to preserve their general capabilities, while all the decomposition matrices within the low-rank matrix are trainable.

# C    MORE EXPERIMENTAL RESULTS

Due to space constraints in the main text, we included only one dataset for each of the four downstream tasks: RTE, WikiQA, Winogrande, and SQuAD. The experimental results for MRPC and QASC are presented in this section.

|  |  | C-Eval | GSM8K | MMLU | HumanEval | Avg. | Performance |
|---|---|---|---|---|---|---|---|
|  | Pre-trained model | 0.4386 | 0.7922 | 0.6594 | 0.5914 | 0.6204 | 0.686 |
|  | Full SFT | 0.2603 | 0.0 | 0.2483 | 0.0 | 0.1271 | 0.887 |
| MRPC | L1-reg (Kirkpatrick et al., 2017) | 0.4062 | 0.7470 | 0.6200 | 0.5434 | 0.5766 | 0.821 |
|  | V-SoftMask (Ke et al., 2023) | 0.4200 | 0.7474 | 0.5229 | 0.5122 | 0.5506 | **0.888** |
|  | CoFiTune (Zhang et al., 2024) | 0.4513 | 0.7863 | 0.6382 | 0.5821 | 0.6145 | 0.884 |
|  | LoRA (Hu et al., 2021) | 0.4546 | 0.7890 | 0.6506 | 0.5936 | **0.6210** | 0.846 |
|  | FAPM (Ours) | 0.4662 | 0.7711 | 0.6410 | 0.5791 | 0.6144 | 0.882 |
|  | Pre-trained model | 0.4386 | 0.7922 | 0.6594 | 0.5914 | 0.6204 | 0.630 |
|  | Full SFT | 0.4284 | 0.0379 | 0.5115 | 0.0121 | 0.2474 | 0.864 |
| QASC | L1-reg (Kirkpatrick et al., 2017) | 0.4133 | 0.7744 | 0.6119 | 0.5507 | 0.5875 | 0.802 |
|  | V-SoftMask (Ke et al., 2023) | 0.4372 | 0.7245 | 0.5922 | 0.5781 | 0.5830 | 0.853 |
|  | CoFiTune (Zhang et al., 2024) | 0.4836 | 0.7919 | 0.6457 | 0.5992 | **0.6301** | 0.835 |
|  | LoRA (Hu et al., 2021) | 0.4833 | 0.7930 | 0.6471 | 0.5731 | 0.6241 | 0.856 |
|  | FAPM (Ours) | 0.4836 | 0.7983 | 0.6326 | 0.5914 | 0.6265 | **0.863** |

Table 5: More results of different CF methods on various datasets using the Llama3-8B model.

|  |  | C-Eval | GSM8K | MMLU | HumanEval | Avg. | Performance |
|---|---|---|---|---|---|---|---|
|  | Pre-trained model | 0.7478 | 0.8180 | 0.6884 | 0.7682 | 0.7556 | 0.765 |
|  | Full SFT | 0.2598 | 0.0 | 0.2481 | 0.0 | 0.1269 | 0.914 |
| MRPC | L1-reg (Kirkpatrick et al., 2017) | 0.7136 | 0.7779 | 0.6261 | 0.7171 | 0.7086 | 0.823 |
|  | V-SoftMask (Ke et al., 2023) | 0.7418 | 0.6933 | 0.6095 | 0.6901 | 0.6836 | **0.919** |
|  | CoFiTune (Zhang et al., 2024) | 0.7612 | 0.8036 | 0.6795 | 0.7317 | 0.7440 | 0.899 |
|  | LoRA (Hu et al., 2021) | 0.7468 | 0.8125 | 0.6873 | 0.7439 | 0.7476 | 0.873 |
|  | FAPM (Ours) | 0.7564 | 0.7938 | 0.6837 | 0.7682 | **0.7505** | 0.892 |
|  | Pre-trained model | 0.7478 | 0.8180 | 0.6884 | 0.7682 | 0.7556 | 0.701 |
|  | Full SFT | 0.5876 | 0.0470 | 0.5445 | 0.2621 | 0.3603 | 0.866 |
| QASC | L1-reg (Kirkpatrick et al., 2017) | 0.7300 | 0.7813 | 0.6453 | 0.7091 | 0.7164 | 0.781 |
|  | V-SoftMask (Ke et al., 2023) | 0.7452 | 0.7636 | 0.6388 | 0.7195 | 0.7167 | **0.857** |
|  | CoFiTune (Zhang et al., 2024) | 0.7744 | 0.8006 | 0.6778 | 0.7500 | 0.7507 | 0.848 |
|  | LoRA (Hu et al., 2021) | 0.7677 | 0.8218 | 0.6872 | 0.7134 | 0.7475 | 0.855 |
|  | FAPM (Ours) | 0.7679 | 0.8157 | 0.6815 | 0.7500 | **0.7538** | 0.851 |

Table 6: More results of different CF methods on various datasets using the Qwen2-7B model.

|  |  | C-Eval | GSM8K | MMLU | HumanEval | Avg. | Performance |
|---|---|---|---|---|---|---|---|
|  | Magnitude (Han et al., 2015) | 0.3801 | 0.6100 | 0.3378 | 0.4731 | 0.4502 | **0.892** |
| MRPC | Wanda (Sun et al., 2023) | 0.4635 | 0.7845 | 0.6506 | 0.5958 | **0.6236** | 0.816 |
|  | FAPM (Ours) | 0.4662 | 0.7711 | 0.6410 | 0.5791 | 0.6144 | 0.882 |
|  | Magnitude (Han et al., 2015) | 0.4916 | 0.7263 | 0.6053 | 0.5223 | 0.5864 | 0.861 |
| QASC | Wanda (Sun et al., 2023) | 0.4705 | 0.7819 | 0.6456 | 0.5886 | 0.6216 | 0.839 |
|  | FAPM (Ours) | 0.4836 | 0.7983 | 0.6326 | 0.5914 | **0.6265** | **0.863** |

Table 7: More results of different pruning methods on various datasets using the Llama3-8B model.

|  |  | C-Eval | GSM8K | MMLU | HumanEval | Avg. | Performance |
|---|---|---|---|---|---|---|---|
|  | Magnitude (Han et al., 2015) | 0.7412 | 0.1296 | 0.2473 | 0.1768 | 0.3238 | **0.911** |
| MRPC | Wanda (Sun et al., 2023) | 0.7458 | 0.7989 | 0.6813 | 0.7482 | 0.7435 | 0.826 |
|  | FAPM (Ours) | 0.7564 | 0.7938 | 0.6837 | 0.7682 | **0.7505** | 0.892 |
|  | Magnitude | 0.7559 | 0.7760 | 0.6407 | 0.7073 | 0.7199 | 0.851 |
| QASC | Wanda (Sun et al., 2023) | 0.7567 | 0.8072 | 0.6858 | 0.7378 | 0.7468 | 0.828 |
|  | FAPM (Ours) | 0.7679 | 0.8157 | 0.6815 | 0.7500 | **0.7538** | 0.851 |

Table 8: More results of different pruning methods on various datasets using the Qwen2-7B model.

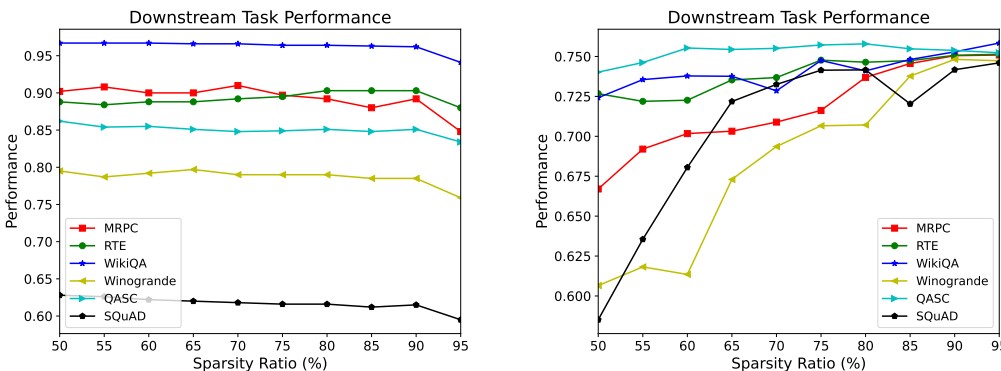

Figure 6: Performance of FAPM on downstream task accuracy and mitigation of catastrophic forgetting with different sparsity ratios on Qwen2-7B.

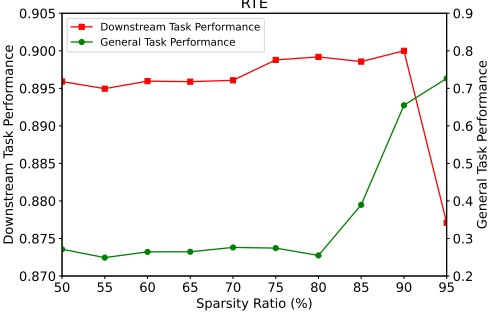

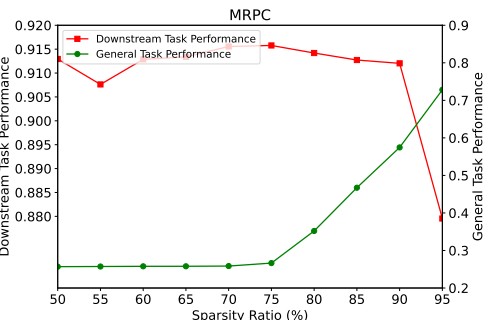

(a) The original accuracy on RTE is 0.890 and the original average accuracy on four general tasks is 0.7556.

(b) The original accuracy on MRPC is 0.914 and the original average accuracy on four general tasks is 0.7556.

Figure 7: The relationship between the magnitude pruning sparsity ratio, general capability, and downstream task performance of Qwen-7B on (a) RTE and (b) MRPC, respectively.

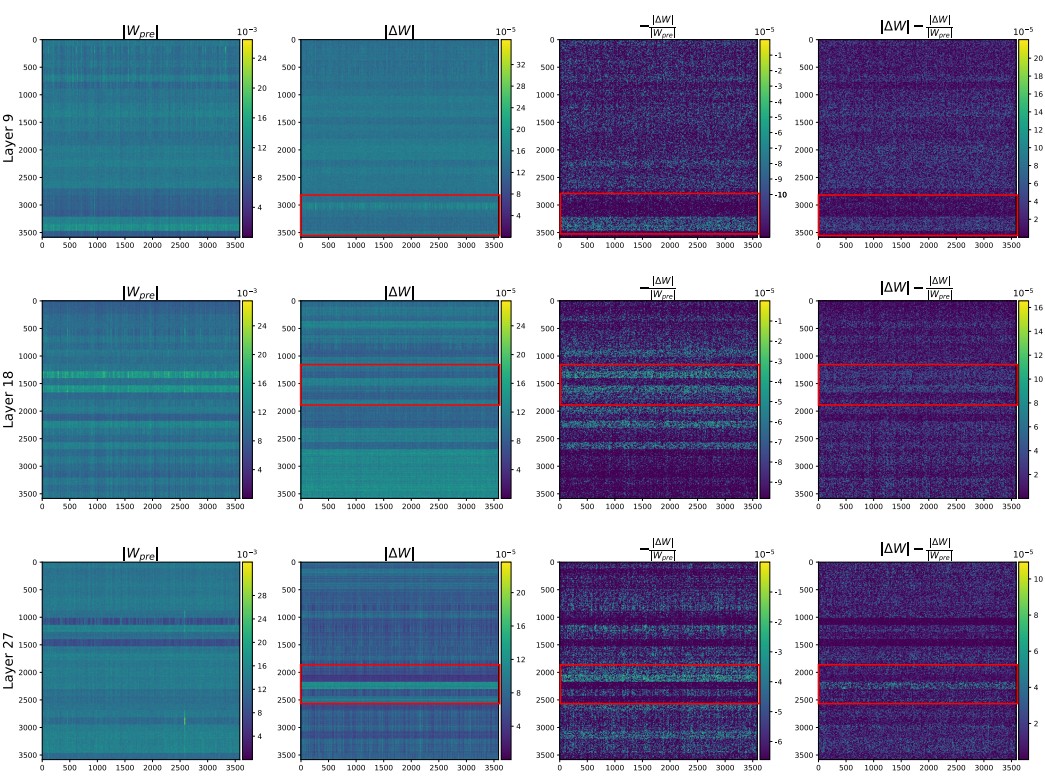

Figure 8: Visualization of the weight matrices in different layers of Qwen2-7B fine-tuned on RTE dataset. From left to right, they represent the magnitude of the pre-trained model weights, the absolute change magnitude of model weights, the relative change magnitude of model weights, and a combination of the absolute and relative change magnitude.

