# OpenReview forum: "Mitigating Catastrophic Forgetting in Large Language Models with Forgetting-aware Pruning"
_ICLR.cc/2025/Conference — ICLR 2025 Conference Withdrawn Submission_

### Official Review · Reviewer_aVtG · 2024-10-16

**Soundness:** 4
**Presentation:** 3
**Contribution:** 3
**Rating:** 6
**Confidence:** 4

**Summary:**

This paper focuses on addressing the issue of catastrophic forgetting (CF) in large language models (LLMs) during fine-tuning, i.e., striking a balance between speciality and generality.
It proposes Forgetting-Aware Pruning Metric (FAPM): A novel method that integrates the ratio of task vector magnitude to pre-trained model parameters into the pruning criteria, allowing for the mitigation of CF without modifying the model architecture or training process.
FAPM successfully mitigates forgetting while retaining 99% of downstream task accuracy across various natural language tasks.
The experiments show that FAPM performs competitively against state-of-the-art methods.

**Strengths:**

1.This paper introduces two key concepts through preliminary experiments—“absolute change magnitude” and “relative change magnitude”—which are designed to balance the trade-off between task specialization and generality.

2.Building on these concepts, the paper proposes the Forgetting-Aware Pruning Metric (FAPM), aimed at deriving an optimal task vector to obtain the final model weights.

3.Extensive experiments demonstrate the effectiveness of FAPM, showing significant improvements over baseline methods.

**Weaknesses:**

1.This paper overlooks an important baseline—Wise-FT [1], a standard model parameter merging method.

2.The experimental setup for the preliminary experiments, particularly those in Sections 2 and 3, is not clearly detailed, e.g., the experimental model, raising concerns about whether similar observations would hold for other models.

3.While instruction-following is a critical aspect of generality, this paper focuses solely on world knowledge and generic reasoning, omitting this aspect.

4.The rationale and motivation for striking a balance between speciality and generality is not clearly articulated in the paper.

[1] Mitchell Wortsman, Gabriel Ilharco, Mike Li, Jong Wook Kim, Hannaneh Hajishirzi, Ali Farhadi, Hongseok Namkoong, and Ludwig Schmidt. 2021. Robust fine-tuning of zero-shot models. 2022 IEEE/CVF Conference on Computer Vision and Pattern Recognition (CVPR), pages 7949–7961.

**Questions:**

1.The limitation mentioned in lines 81-83, stating that "methods that alter the training process or model architecture not only make the training process more difficult to control but also degrade the accuracy of downstream tasks," lacks references to prior work supporting this claim.

2.The figures in this paper appear not to be in vector graphic format, which may affect the clarity and quality of the visualizations.

3.The quotation marks in line 212 are not properly formatted according to standard usage.

---

> ### Author Response · Authors · 2024-11-18
> **To Reviewer aVtG**
>
> Thank you for Reviewer YFJm's review.
>
> # Q1: This paper overlooks an important baseline—Wise-FT [1], a standard model parameter merging method.
>
> We appreciate the reviewer for providing an important baseline. We have added a comparison between Wise-FT and our method. The original Wise-FT paper mentions: "WiSE-FT using a fixed mixing coefficient α=0.5 with the fixed optimal mixing coefficient." Therefore, we adhere to the hyperparameter settings as described in the original paper. The experimental results can be seen in the following two tables.
> | Tasks      | Methods            | C-Eval  | GSM8K  | MMLU   | HumanEval | Avg.   | Performance |
> |:----------:|:------------------:|:-------:|:------:|:------:|:---------:|:------:|:-----------:|
> | RTE        | Pre-trained model  | 0.4386  | 0.7922 | 0.6594 | 0.5914    | 0.6204 | 0.819       |
> | RTE        | Full SFT           | 0.2311  | 0.075  | 0.2554 | 0.0       | 0.1403 | 0.890       |
> | RTE        | WiSE-FT            | 0.3046  | 0.4420 | 0.4255 | 0.5609    | 0.4332 | 0.889       |
> | RTE        | FAPM(Ours)         | 0.4623  | 0.7915 | 0.6454 | 0.5975    | 0.6242 | 0.897       |
> |||||||||
> | WikiQA     | Pre-trained model  | 0.4386  | 0.7922 | 0.6594 | 0.5914    | 0.6204 | 0.913       |
> | WikiQA     | Full SFT           | 0.2547  | 0.0    | 0.2422 | 0.0       | 0.1242 | 0.966       |
> | WikiQA     | WiSE-FT            | 0.2581  | 0.0    | 0.2458 | 0.0       | 0.1259 | 0.958       |
> | WikiQA     | FAPM(Ours)         | 0.4749  | 0.7975 | 0.6563 | 0.5853    | 0.6285 | 0.964       |
> |||||||||
> | Winogrande | Pre-trained model  | 0.4386  | 0.7922 | 0.6594 | 0.5914    | 0.6204 | 0.519       |
> | Winogrande | Full SFT           | 0.2792  | 0.0606 | 0.3438 | 0.0       | 0.1709 | 0.820       |
> | Winogrande | WiSE-FT            | 0.4741  | 0.4412 | 0.5967 | 0.5060    | 0.5045 | 0.830       |
> | Winogrande | FAPM(Ours)         | 0.4829  | 0.7680 | 0.6472 | 0.5731    | 0.6178 | 0.824       |
> |||||||||
> | SQuAD      | Pre-trained model  | 0.4386  | 0.7922 | 0.6594 | 0.5914    | 0.6204 | 0.371       |
> | SQuAD      | Full SFT           | 0.2806  | 0.0212 | 0.3206 | 0.0       | 0.1556 | 0.646       |
> | SQuAD      | WiSE-FT            | 0.4309  | 0.4009 | 0.5484 | 0.5102    | 0.4725 | 0.639       |
> | SQuAD      | FAPM(Ours)         | 0.4738  | 0.7310 | 0.6455 | 0.5748    | 0.6063 | 0.637       |
>
> Table 1: Comparison of FAPM and WiSE-FT on the Llama3-8B model. The 'Avg.' column represents the performance on general tasks, while the 'Performance' column indicates the performance on downstream tasks.
>
> | Tasks      | Methods            | C-Eval  | GSM8K   | MMLU   | HumanEval | Avg.   | Performance |
> |:----------:|:------------------:|:-------:|:-------:|:------:|:---------:|:------:|:-----------:|
> | RTE        | Pre-trained model  | 0.7478  | 0.8180  | 0.6884 | 0.7682    | 0.7556 | 0.574       |
> | RTE        | Full SFT           | 0.2602  | 0.075   | 0.2423 | 0.0       | 0.1443 | 0.890       |
> | RTE        | WiSE-FT            | 0.7488  | 0.7664  | 0.4638 | 0.7682    | 0.6868 | 0.888       |
> | RTE        | FAPM(Ours)         | 0.7568  | 0.8104  | 0.6857 | 0.7500    | 0.7507 | 0.903       |
> |||||||||
> | WikiQA     | Pre-trained model  | 0.7478  | 0.8180  | 0.6884 | 0.7682    | 0.7556 | 0.896       |
> | WikiQA     | Full SFT           | 0.2510  | 0.076   | 0.2434 | 0.0       | 0.1426 | 0.965       |
> | WikiQA     | WiSE-FT            | 0.2476  | 0.0     | 0.2414 | 0.0       | 0.1222 | 0.963       |
> | WikiQA     | FAPM(Ours)         | 0.7555  | 0.8036  | 0.6902 | 0.7621    | 0.7529 | 0.962       |
> |||||||||
> | Winogrande | Pre-trained model  | 0.7478  | 0.8180  | 0.6884 | 0.7682    | 0.7556 | 0.558       |
> | Winogrande | Full SFT           | 0.4090  | 0.0303  | 0.2996 | 0.0609    | 0.1999 | 0.790       |
> | Winogrande | WiSE-FT            | 0.6747  | 0.5343  | 0.5821 | 0.5914    | 0.5956 | 0.780       |
> | Winogrande | FAPM(Ours)         | 0.7618  | 0.8068  | 0.6845 | 0.7395    | 0.7482 | 0.785       |
> |||||||||
> | SQuAD      | Pre-trained model  | 0.7478  | 0.8180  | 0.6884 | 0.7682    | 0.7556 | 0.451       |
> | SQuAD      | Full SFT           | 0.3531  | 0.02122 | 0.3183 | 0.0       | 0.1731 | 0.624       |
> | SQuAD      | WiSE-FT            | 0.6784  | 0.5743  | 0.5868 | 0.5524    | 0.5979 | 0.622       |
> | SQuAD      | FAPM(Ours)         | 0.7410  | 0.8006  | 0.6752 | 0.7500    | 0.7417 | 0.615       |
>
> Table 2: Comparison of FAPM and WiSE-FT on the Qwen2-7B.
>
> From the two tables above, we can conclude that WiSE-FT achieves comparable downstream task accuracy to FAPM. However, WiSE-FT still suffers from significant catastrophic forgetting.

---

> ### Author Response · Authors · 2024-11-18
> **To Reviewer aVtG (Continue)**
>
> # Q2: The experimental setup for the preliminary experiments, particularly those in Sections 2 and 3, is not clearly detailed, e.g., the experimental model, raising concerns about whether similar observations would hold for other models.
>
> Following the reviewer's feedback on Sections 2 and 3 of the paper, we have provided explanations for the experimental settings in Figures 2 and 3. The revised paper has been re-uploaded. In Figures 2 and 3, the model we used is Llama3-8B. To further illustrate our observations, we conducted experiments on Qwen2-7B and included the related figures (Figures 7 and 8) in the appendix, which can be found in lines 843 - 900. From Figures 7 and 8, we observed similar patterns on Qwen2-7B as those on Llama3-8B.
>
> # Q3: While instruction-following is a critical aspect of generality, this paper focuses solely on world knowledge and generic reasoning, omitting this aspect
>
> We introduced two domain-specific instruction-following scenarios, namely the medical domain and the mathematical domain. For medical data, we chose the MedMCQA dataset, and for mathematical data, we selected the MetaMathQA dataset. We validated the effectiveness of our method on these two scenarios using Llama3-8B and Qwen2-7B models. The experimental results can be seen in the following two tables. It is worth mentioning that our method remains effective even when catastrophic forgetting occurs during LoRA fine-tuning. Specific results can be found in our response to Reviewer YFJm.
> | Tasks      | Methods            | C-Eval  | GSM8K  | MMLU   | HumanEval | Avg.   | Performance |
> |:----------:|:------------------:|:-------:|:------:|:------:|:---------:|:------:|:-----------:|
> | MedMCQA    | Pre-trained model  | 0.4386  | 0.7922 | 0.6594 | 0.5914    | 0.6204 | 0.5169      |
> | MedMCQA    | Full SFT           | 0.3315  | 0.4897 | 0.5365 | 0.4434    | 0.4502 | 0.5862      |
> | MedMCQA    | FAPM(Ours)         | 0.4586  | 0.7733 | 0.6638 | 0.5731    | 0.6172 | 0.5935      |
> |||||||||
> | MetaMathQA | Pre-trained model  | 0.4386  |      - | 0.6594 | 0.5914    | 0.5631 | 0.7922      |
> | MetaMathQA | Full SFT           | 0.3101  |      - | 0.4411 | 0.1890    | 0.3133 | 0.8178      |
> | MetaMathQA | FAPM(Ours)         | 0.4697  |      _ | 0.6447 | 0.5729    | 0.5624 | 0.8121      |
>
> Table 3: The results of FAPM on various datasets using Llama3-8B. The 'Avg.' column represents the performance on general tasks, while the 'Performance' column indicates the performance on downstream tasks. Since the downstream task is a mathematical task, we used gsm8k for testing. Therefore, when calculating the performance of general tasks, we did not include gsm8k, which is indicated by a '-'.
>
> | Tasks      | Methods            | C-Eval  | GSM8K  | MMLU   | HumanEval | Avg.   | Performance |
> |:----------:|:------------------:|:-------:|:------:|:------:|:---------:|:------:|:-----------:|
> | MedMCQA    | Pre-trained model  | 0.7478  | 0.8180 | 0.6884 | 0.7682    | 0.7556 | 0.5208      |
> | MedMCQA    | Full SFT           | 0.5662  | 0.3731 | 0.5001 | 0.6443    | 0.5209 | 0.5811      |
> | MedMCQA    | FAPM(Ours)         | 0.7613  | 0.7968 | 0.6732 | 0.7445    | 0.7439 | 0.5713      |
> |||||||||
> | MetaMathQA | Pre-trained model  | 0.7478  |      - | 0.6884 | 0.7682    | 0.7347 | 0.8180      |
> | MetaMathQA | Full SFT           | 0.5582  |      - | 0.4319 | 0.3242    | 0.4381 | 0.8499      |
> | MetaMathQA | FAPM(Ours)         | 0.7468  |      - | 0.6857 | 0.7511    | 0.7278 | 0.8484      |
>
> Table 4: The results of FAPM on various datasets using Qwen2-7B.
>
> From the table above, we can see that our method still effectively alleviates forgetting to a mere 1% while maintaining an impressive 99% accuracy on downstream tasks in both the mathematical and medical task scenarios.

---

> ### Author Response · Authors · 2024-11-18
> **To Reviewer aVtG (Continue)**
>
> # Q4: The rationale and motivation for striking a balance between speciality and generality is not clearly articulated in the paper.
>
> The issue addressed by catastrophic forgetting research is how to enhance a model’s performance on downstream specialized tasks (speciality) without compromising its effectiveness on general tasks (generality). Existing methods [1][2][3] that tackle catastrophic forgetting maintain the model's general performance but may, to varying degrees, impair its performance on specialized tasks. Therefore, these methods usually aim to find a balance between speciality and generality. Our approach addresses the same issue as these methods, and thus our goal is also striking a balance between speciality and generality. We believe this goal is equivalent to “mitigating catastrophic forgetting,” and therefore, we use these two expressions interchangeably in our paper.
>
> # Q5: The limitation mentioned in lines 81-83, stating that "methods that alter the training process or model architecture not only make the training process more difficult to control but also degrade the accuracy of downstream tasks," lacks references to prior work supporting this claim.
>
> We have provided references [1][4] to support the claims in our paper. See lines 81 - 83. The experimental results from these two papers demonstrate that modifying the training process (e.g., V-SoftMask [4], HAT-Adapter [5], etc.) or altering the model architecture (e.g., LoRA) can degrade the accuracy of downstream tasks. Refer to Table 2 in [1] and Table 2 in [4] for more details.
>
> [1] Hengyuan Zhang, Yanru Wu, Dawei Li, Zacc Yang, Rui Zhao, Yong Jiang, and Fei Tan. Balancing
> speciality and versatility: a coarse to fine framework for supervised fine-tuning large language
> model. arXiv preprint arXiv:2404.10306, 2024.
>
> [2] Jianheng Huang, Leyang Cui, Ante Wang, Chengyi Yang, Xinting Liao, Linfeng Song, Junfeng Yao,
> and Jinsong Su. Mitigating catastrophic forgetting in large language models with self-synthesized
> rehearsal. arXiv preprint arXiv:2403.01244, 2024.
>
> [3] Yajing Kong, Liu Liu, Huanhuan Chen, Janusz Kacprzyk, and Dacheng Tao. Overcoming catas-
> trophic forgetting in continual learning by exploring eigenvalues of hessian matrix. IEEE Trans-
> actions on Neural Networks and Learning Systems, 2023.
>
> [4] Ke, Z., Shao, Y., Lin, H., Konishi, T., Kim, G., & Liu, B. Continual Pre-training of Language Models. In The Eleventh International Conference on Learning Representations.
>
> [5] Zixuan Ke, Hu Xu, and Bing Liu. Adapting bert for continual learning of a sequence of aspect sentiment classification tasks. In NAACL, pp. 4746–4755, 2021c.
>
> # Q6: The figures in this paper appear not to be in vector graphic format, which may affect the clarity and quality of the visualizations.
>
> Thank you for your thorough review. To enable readers to view the images in our paper more clearly, we have replaced the images in the original paper from PNG format to vector graphic format. The revised paper has been re-uploaded.
>
> # Q7: The quotation marks in line 212 are not properly formatted according to standard usage.
>
> We appreciate the diligent reviewers for pointing out the improperly formatted quotation marks in our paper. We have made the necessary corrections.

---

> > ### Comment · Reviewer_aVtG · 2024-11-22
> > **Response to Author**
> >
> > The author have well addressed my concerns, I am happy to modify the Soundness score.

---

### Official Review · Reviewer_53VB · 2024-11-01

**Soundness:** 2
**Presentation:** 1
**Contribution:** 1
**Rating:** 3
**Confidence:** 4

**Summary:**

The authors identified pruning as a way to mitigate catastrophic forgetting. Specifically, the authors propose to prune weight updates due to fine-tuning, using a combination of pruning metrics including absolute magnitude of change in weights and relative magnitude of change in weights relative to the magnitude of pre-trained weights.

**Strengths:**

- Using pruning to mitigate catastrophic forgetting during fine-tuning is novel.
- This paper contains some original insights about the generality/specificity trade-off when fine-tuning LLMs.

**Weaknesses:**

- This paper proposes a complicated pruning metrics that only shows marginal improvement over prior art.
- The quality of write-up is generally poor. For example, section 3.1 is entirely unnecessary and can be in the appendix without affecting the flow of the paper.
-  No ablation study. Why not use the relative portion of the pruning metric alone? It’s a complicated scoring metric, please ablate it.

**Questions:**

- See weakness.

---

> ### Author Response · Authors · 2024-11-13
> **To Reviewer 53VB**
>
> Thank you for Reviewer 53VB's review.
>
>
> # Q1: This paper proposes a complicated pruning metrics that only shows marginal improvement over prior art.
>
> 1. We cannot agree with your comment of "a complicated pruning metrics." This is because our method is sufficiently simple. FAPM only requires pruning the incremental model after fine-tuning, without necessitating any changes to the model training process, and our pruning procedure is also simple enough. To visually demonstrate the simplicity of FAPM, we have provided the PyTorch implementation code for the FAPM pruning metrics below. The formula (1) in the paper can be implemented in just a few lines of code.
>
> ```
> def FAPM(W_pre, W_tv, key):
>     """
>     Computes the FAPM for a specific parameter in a model.
>
>     Parameters:
>     -----------
>     W_pre : OrderedDict
>         The model state_dict of the pre-trained model.
>
>     W_tv : OrderedDict
>         The model state_dict of the task vectors,
>
>     key : str
>         The identifier to specify which parameter matrix to use from both `W_pre` and `W_tv`.
>
>     Returns:
>     --------
>     score : torch.Tensor
>         The importance score (FAPM) for the specified parameter.
>     """
>
>     # Compute the absolute value of the task vector for the specified parameter
>     abs_W_tv = W_tv[key].abs()
>
>     # Compute the mean of the absolute values of the pre-trained weights for the specified parameter
>     mean_abs_W_pre = W_pre[key].abs().mean()
>
>     # Compute the normalized absolute task vector
>     normalized_abs_W_tv = abs_W_tv / W_pre[key].abs()
>
>     # Compute the importance score (FAPM)
>     score = abs_W_tv - mean_abs_W_pre * normalized_abs_W_tv
>
>     return score
> ```
>
> 2. The improvements brought by our method are significant. FAPM effectively alleviates forgetting to a mere 1% while maintaining an impressive 99% accuracy on downstream tasks. Most importantly, compared to previous state-of-the-art methods, our greatest advantage lies in the simplicity of our approach. It does not require any changes to the model training process, does not necessitate adding extra model structures, and does not introduce additional data. Moreover, our method achieves results that are better than or comparable to the previous state-of-the-art methods (refer to Table 1 and Table 2 in the paper).
>
> 3. The simplicity and effectiveness of our method have been fully recognized by other reviewers, e.g., *"The simplicity of the technique introduced by the authors is really great"* commented  by Reviewer YFJm and *"FAPM doesn’t complicate the model architecture or training process, making it easier to implement"* commented by Reviewer HmBx. Therefore, we believe that Reviewer 53VB's statement regarding our method as "a complicated pruning metric that only shows marginal improvement over prior art" is not appropriate. We cannot agree with this assessment.
>
> # Q2: The quality of write-up is generally poor. For example, section 3.1 is entirely unnecessary and can be in the appendix without affecting the flow of the paper.
>
> We do not believe that Section 3.1 is unnecessary because this section presents our analysis of the causes of the CF problem, elucidates our perspective on the issue, and introduces the starting point of our method. We believe these analyses are necessary and insightful. As mentioned by Reviewer YFJm, *"I loved reading section 3.1 wherein they provide analysis and a clear thinking process of how they arrived at the method"*, this section helps readers to better understand our motivation.

---

> ### Author Response · Authors · 2024-11-13
> **To Reviewer 53VB (Continue)**
>
> Thank you for Reviewer 53VB's review.
>
> #  Q3: Why not use the relative portion of the pruning metric alone? It’s a complicated scoring metric, please ablate it.
>
> In the paper, we presented comparative experiments with the pruning method that uses only “absolute change magnitude” (refer to Tables 3 and 4 in the paper). Here, we supplement the results of ablation experiments that use only “relative change magnitude” (only RCM) for pruning. We conducted the experiments on Llama3-8B, with the pruning ratio consistent with that in the paper, which is 90%.
>
>
> | Tasks| Methods         | ceval   | gsm8k  | mmlu   | humaneval | Avg.   | Performance |
> |-----|---------|---------|--------|--------|-----------|--------|-------------|
> |RTE | Pre-trained model | 0.4386  | 0.7922 | 0.6594 | 0.5914    | 0.6204 | 0.819       |
> |RTE | Full SFT          | 0.2311  | 0.075  | 0.2554 | 0.0       | 0.1403 | 0.890       |
> |RTE |only RCM          | 0.4391  | 0.7915 | 0.6620 | 0.6097    | 0.6255 | 0.823       |
> |RTE |FAPM              | 0.4623  | 0.7915 | 0.6454 | 0.5975    | 0.6242 | 0.897       |
> |WikiQA | Pre-trained model | 0.4386  | 0.7922 | 0.6594 | 0.5914    | 0.6204 | 0.913       |
> |WikiQA |Full SFT          | 0.2547  | 0.0    | 0.2422 | 0.0       | 0.1242 | 0.966       |
> |WikiQA |only RCM          | 0.4385  | 0.7892 | 0.6589 | 0.6097    | 0.6240 | 0.947       |
> |WikiQA |FAPM              | 0.4749  | 0.7975 | 0.6563 | 0.5853    | 0.6285 | 0.964       |
> |Winogrande | Pre-trained model | 0.4386  | 0.7922 | 0.6594 | 0.5914    | 0.6204 | 0.519       |
> |Winogrande |Full SFT          | 0.2792  | 0.0606 | 0.3438 | 0.0       | 0.1709 | 0.820       |
> |Winogrande |only RCM          | 0.4369  | 0.7899 | 0.6597 | 0.5975    | 0.6209 | 0.551       |
> |Winogrande |FAPM              | 0.4829  | 0.7680 | 0.6472 | 0.5731    | 0.6178 | 0.824       |
> |SQuAD | Pre-trained model | 0.4386  | 0.7922 | 0.6594 | 0.5914    | 0.6204 | 0.371       |
> |SQuAD |Full SFT          | 0.2806  | 0.0212 | 0.3206 | 0.0       | 0.1556 | 0.646       |
> |SQuAD |only RCM          | 0.4449  | 0.7915 | 0.6582 | 0.5975    | 0.6230 | 0.374       |
> |SQuAD |FAPM              | 0.4738  | 0.7310 | 0.6455 | 0.5748    | 0.6063 | 0.637       |
>
> Table 1: The results of using only RCM on the Llama3-8B model. The 'Avg.' column represents the average performance on general tasks, while the 'Performance' column indicates the performance on downstream tasks.
>
> From the table, it can be seen that using only the relative change magnitude cannot ensure the performance on downstream tasks; the model performance is comparable to that of the pre-trained model. The reason is that pruning using the “relative change magnitude” is designed solely to address catastrophic forgetting. Using only the relative change magnitude results in retaining parameter values that are relatively small in the vector during pruning, which cannot guarantee the performance on downstream tasks.

---

> ### Author Response · Authors · 2024-11-25
> **Hope to communicate with reviewer 53VB regarding the paper.**
>
> Dear Reviewer 53VB,
>
> Thank you once again for your valuable feedback. We have submitted our responses to your comments and greatly appreciate your insights. As the discussion phase between the reviewer and the authors approaches its deadline, we kindly request that you share any further thoughts or questions at your earliest convenience.
>
> Best regards, Authors

---

### Official Review · Reviewer_HmBx · 2024-11-03

**Soundness:** 3
**Presentation:** 3
**Contribution:** 2
**Rating:** 5
**Confidence:** 3

**Summary:**

In this paper, the authors introduce Forgetting-Aware Pruning Metric (FAPM), a new pruning-based method aimed at striking a balance between reducing forgetting and maintaining downstream task performance. They find that the overlap between task vectors – which represent the difference between pre-trained weights and those fine-tuned on new tasks – and pre-trained model parameters is a major factor in forgetting. FAPM uses this overlap ratio as a metric for assessing forgetting and incorporates it into pruning decisions. A key advantage is that FAPM doesn't require changes to the training process or the model architecture, nor does it need any additional data. The authors conducted experiments across six datasets covering various tasks, and the results show that FAPM keeps forgetting to just 1% while achieving 99% accuracy on downstream tasks, making it very competitive against other state-of-the-art methods.

**Strengths:**

1. FAPM effectively limits forgetting to a mere 1%, which is a significant improvement.
2. The method maintains an impressive 99% accuracy in downstream tasks, showing that it preserves performance.
3. Unlike some existing methods, FAPM doesn’t complicate the model architecture or training process, making it easier to implement.
4. This approach doesn’t rely on replaying original training data or using additional datasets, making it more practical in various scenarios.

**Weaknesses:**

1. The reliance on the overlap between task vectors and pre-trained weights might not generalize across all types of tasks or models.
2. While pruning is effective, it may not address all aspects of learning or forgetting, so there could be other methods worth exploring along with FAPM.
3. While the experiments are comprehensive, testing on even more diverse datasets could strengthen the generalizability of the findings.
4. The paper doesn’t discuss how FAPM performs in highly specialized or unusual task scenarios, which might reveal limitations.

**Questions:**

1. This appears to be a unique model merging method; however, why is there no comparison made with other methods of a similar nature?
2. While you have elucidated how to control FAPM to achieve a better balance between accuracy and forgetting levels, is there a more effective approach to directly estimate an optimal balance point rather than adjusting parameters through multiple rounds of performance feedback?

---

> ### Author Response · Authors · 2024-11-15
> **To Reviewer HmBx**
>
> Thank you for Reviewer HmBx's review.
>
> # Response to Weaknesses
>
> The four issues raised by reviewer HmBx in the weakness section primarily concern the generalizability of our proposed method or its potential limitations when applied to more specialized scenarios. To better demonstrate the generalizability of our method, we introduced two more specialized scenarios: a medical scenario and a mathematical scenario. For medical data, we selected the MedMCQA dataset, and for mathematical data, we selected the MetaMathQA dataset. We validated the effectiveness of our method in these two scenarios using Llama3-8B and Qwen2-7B. The experimental results are presented in the following two tables. Notably, even when catastrophic forgetting occurs during LoRA fine-tuning, our method remains effective. The detailed results can be found in our response to reviewer YFJm.
>
> | Tasks      | Methods            | C-Eval  | GSM8K  | MMLU   | HumanEval | Avg.   | Performance |
> |:----------:|:------------------:|:-------:|:------:|:------:|:---------:|:------:|:-----------:|
> | MedMCQA    | Pre-trained model  | 0.4386  | 0.7922 | 0.6594 | 0.5914    | 0.6204 | 0.5169      |
> | MedMCQA    | Full SFT           | 0.3315  | 0.4897 | 0.5365 | 0.4434    | 0.4502 | 0.5862      |
> | MedMCQA    | FAPM(Ours)         | 0.4586  | 0.7733 | 0.6638 | 0.5731    | 0.6172 | 0.5935      |
> |||||||||
> | MetaMathQA | Pre-trained model  | 0.4386  |      - | 0.6594 | 0.5914    | 0.5631 | 0.7922      |
> | MetaMathQA | Full SFT           | 0.3101  |      - | 0.4411 | 0.1890    | 0.3133 | 0.8178      |
> | MetaMathQA | FAPM(Ours)         | 0.4697  |      _ | 0.6447 | 0.5729    | 0.5624 | 0.8121      |
>
> Table 1: The results of FAPM on various datasets using Llama3-8B. The 'Avg.' column represents the average performance on general tasks, while the 'Performance' column indicates the performance on downstream tasks. Since the downstream task is a mathematical task, we used gsm8k for testing. Therefore, when calculating the performance of general tasks, we did not include gsm8k, which is indicated by a '-'.
>
>
> | Tasks      | Methods            | C-Eval  | GSM8K  | MMLU   | HumanEval | Avg.   | Performance |
> |:----------:|:------------------:|:-------:|:------:|:------:|:---------:|:------:|:-----------:|
> | MedMCQA    | Pre-trained model  | 0.7478  | 0.8180 | 0.6884 | 0.7682    | 0.7556 | 0.5208      |
> | MedMCQA    | Full SFT           | 0.5662  | 0.3731 | 0.5001 | 0.6443    | 0.5209 | 0.5811      |
> | MedMCQA    | FAPM(Ours)         | 0.7613  | 0.7968 | 0.6732 | 0.7445    | 0.7439 | 0.5713      |
> |||||||||
> | MetaMathQA | Pre-trained model  | 0.7478  |      - | 0.6884 | 0.7682    | 0.7347 | 0.8180      |
> | MetaMathQA | Full SFT           | 0.5582  |      - | 0.4319 | 0.3242    | 0.4381 | 0.8499      |
> | MetaMathQA | FAPM(Ours)         | 0.7468  |      - | 0.6857 | 0.7511    | 0.7278 | 0.8484      |
>
> Table 2: The results of FAPM on various datasets using Qwen2-7B.
>
> From the table above, we can see that our method still effectively alleviates forgetting to a mere 1% while maintaining an impressive 99% accuracy on downstream tasks in both the mathematical and medical task scenarios.

---

> > ### Comment · Reviewer_HmBx · 2024-11-25
> >
> > Thank you to the author for the patient response. I have carefully reviewed the replies from the other reviewers. I maintain my score unchanged.

---

> > > ### Author Response · Authors · 2024-11-25
> > > **To review HmBx**
> > >
> > > We sincerely appreciate your response.
> > >
> > > We have carefully read your review, which raises the same concerns as Reviewer aVtG. The main issues are our method's generalization ability and the baseline without model merging. In their subsequent response, Reviewer aVtG mentioned, "The author has well addressed my concerns; I am happy to modify the Soundness score." We are unsure if our response and experiments have resolved your initial concerns about the paper. This would help us further improve the quality of the paper.

---

> ### Author Response · Authors · 2024-11-15
> **To Reviewer HmBx (Continue)**
>
> Thank you for Reviewer HmBx's review.
>
> # Q2. This appears to be a unique model merging method; however, why is there no comparison made with other methods of a similar nature?
>
> Reviewer aVtG also mentioned this issue, stating, "This paper overlooks an important baseline—Wise-FT [1], a standard model parameter merging method," and provided a baseline for model merging for comparison. The original Wise-FT paper notes: "WiSE-FT using a fixed mixing coefficient α=0.5 with the fixed optimal mixing coefficient." Therefore, we followed the hyperparameter settings from the original paper. The experimental results are presented in the following two tables.
>
> | Tasks      | Methods            | C-Eval  | GSM8K  | MMLU   | HumanEval | Avg.   | Performance |
> |:----------:|:------------------:|:-------:|:------:|:------:|:---------:|:------:|:-----------:|
> | RTE        | Pre-trained model  | 0.4386  | 0.7922 | 0.6594 | 0.5914    | 0.6204 | 0.819       |
> | RTE        | Full SFT           | 0.2311  | 0.075  | 0.2554 | 0.0       | 0.1403 | 0.890       |
> | RTE        | WiSE-FT            | 0.3046  | 0.4420 | 0.4255 | 0.5609    | 0.4332 | 0.889       |
> | RTE        | FAPM(Ours)         | 0.4623  | 0.7915 | 0.6454 | 0.5975    | 0.6242 | 0.897       |
> |||||||||
> | WikiQA     | Pre-trained model  | 0.4386  | 0.7922 | 0.6594 | 0.5914    | 0.6204 | 0.913       |
> | WikiQA     | Full SFT           | 0.2547  | 0.0    | 0.2422 | 0.0       | 0.1242 | 0.966       |
> | WikiQA     | WiSE-FT            | 0.2581  | 0.0    | 0.2458 | 0.0       | 0.1259 | 0.958       |
> | WikiQA     | FAPM(Ours)         | 0.4749  | 0.7975 | 0.6563 | 0.5853    | 0.6285 | 0.964       |
> |||||||||
> | Winogrande | Pre-trained model  | 0.4386  | 0.7922 | 0.6594 | 0.5914    | 0.6204 | 0.519       |
> | Winogrande | Full SFT           | 0.2792  | 0.0606 | 0.3438 | 0.0       | 0.1709 | 0.820       |
> | Winogrande | WiSE-FT            | 0.4741  | 0.4412 | 0.5967 | 0.5060    | 0.5045 | 0.830       |
> | Winogrande | FAPM(Ours)         | 0.4829  | 0.7680 | 0.6472 | 0.5731    | 0.6178 | 0.824       |
> |||||||||
> | SQuAD      | Pre-trained model  | 0.4386  | 0.7922 | 0.6594 | 0.5914    | 0.6204 | 0.371       |
> | SQuAD      | Full SFT           | 0.2806  | 0.0212 | 0.3206 | 0.0       | 0.1556 | 0.646       |
> | SQuAD      | WiSE-FT            | 0.4309  | 0.4009 | 0.5484 | 0.5102    | 0.4725 | 0.639       |
> | SQuAD      | FAPM(Ours)         | 0.4738  | 0.7310 | 0.6455 | 0.5748    | 0.6063 | 0.637       |
>
> Table 3: Comparison of FAPM and WiSE-FT on the Llama3-8B model. The 'Avg.' column represents the average performance on general tasks, while the 'Performance' column indicates the performance on downstream tasks.
>
> | Tasks      | Methods            | C-Eval  | GSM8K   | MMLU   | HumanEval | Avg.   | Performance |
> |:----------:|:------------------:|:-------:|:-------:|:------:|:---------:|:------:|:-----------:|
> | RTE        | Pre-trained model  | 0.7478  | 0.8180  | 0.6884 | 0.7682    | 0.7556 | 0.574       |
> | RTE        | Full SFT           | 0.2602  | 0.075   | 0.2423 | 0.0       | 0.1443 | 0.890       |
> | RTE        | WiSE-FT            | 0.7488  | 0.7664  | 0.4638 | 0.7682    | 0.6868 | 0.888       |
> | RTE        | FAPM(Ours)         | 0.7568  | 0.8104  | 0.6857 | 0.7500    | 0.7507 | 0.903       |
> |||||||||
> | WikiQA     | Pre-trained model  | 0.7478  | 0.8180  | 0.6884 | 0.7682    | 0.7556 | 0.896       |
> | WikiQA     | Full SFT           | 0.2510  | 0.076   | 0.2434 | 0.0       | 0.1426 | 0.965       |
> | WikiQA     | WiSE-FT            | 0.2476  | 0.0     | 0.2414 | 0.0       | 0.1222 | 0.963       |
> | WikiQA     | FAPM(Ours)         | 0.7555  | 0.8036  | 0.6902 | 0.7621    | 0.7529 | 0.962       |
> |||||||||
> | Winogrande | Pre-trained model  | 0.7478  | 0.8180  | 0.6884 | 0.7682    | 0.7556 | 0.558       |
> | Winogrande | Full SFT           | 0.4090  | 0.0303  | 0.2996 | 0.0609    | 0.1999 | 0.790       |
> | Winogrande | WiSE-FT            | 0.6747  | 0.5343  | 0.5821 | 0.5914    | 0.5956 | 0.780       |
> | Winogrande | FAPM(Ours)         | 0.7618  | 0.8068  | 0.6845 | 0.7395    | 0.7482 | 0.785       |
> |||||||||
> | SQuAD      | Pre-trained model  | 0.7478  | 0.8180  | 0.6884 | 0.7682    | 0.7556 | 0.451       |
> | SQuAD      | Full SFT           | 0.3531  | 0.02122 | 0.3183 | 0.0       | 0.1731 | 0.624       |
> | SQuAD      | WiSE-FT            | 0.6784  | 0.5743  | 0.5868 | 0.5524    | 0.5979 | 0.622       |
> | SQuAD      | FAPM(Ours)         | 0.7410  | 0.8006  | 0.6752 | 0.7500    | 0.7417 | 0.615       |
>
> Table 4: Comparison of FAPM and WiSE-FT on the Qwen2-7B.
>
> From the two tables above, we can conclude that WiSE-FT achieves comparable downstream task accuracy to FAPM. However, WiSE-FT still suffers from significant catastrophic forgetting.

---

> ### Author Response · Authors · 2024-11-15
> **To Reviewer HmBx (Continue)**
>
> # Q3. While you have elucidated how to control FAPM to achieve a better balance between accuracy and forgetting levels, is there a more effective approach to directly estimate an optimal balance point rather than adjusting parameters through multiple rounds of performance feedback?
>
> FAPM does not require multiple rounds of performance feedback to achieve a better balance between accuracy and forgetting levels. Instead, it directly controls the balance by calculating the mean of the absolute values of the pre-trained parameter matrix, as shown in Equation 1 by $| W_{pre}^{i} |$. For more details, refer to the textual description of Equation 1 (lines 257 - 272). If the "multiple rounds of performance feedback" you mentioned refer to the *for* loop in our pseudocode or the $i$ in Equation 1, this actually represents the application of our method to each module in the parameter matrix separately (or can be understood as layer-wise operations), rather than adjusting parameters through multiple rounds of performance feedback.

---

> ### Author Response · Authors · 2024-11-25
> **Hope to communicate with reviewer HmBx regarding the paper.**
>
> Dear Reviewer HmBx,
>
> Thank you once again for your valuable feedback. We have submitted our responses to your comments and greatly appreciate your insights. As the discussion phase between the reviewer and the authors approaches its deadline, we kindly request that you share any further thoughts or questions at your earliest convenience.
>
> Best regards, Authors

---

### Official Review · Reviewer_YFJm · 2024-11-04

**Soundness:** 3
**Presentation:** 3
**Contribution:** 3
**Rating:** 6
**Confidence:** 4

**Summary:**

This paper tries to tackle the problem of whether catastrophic forgetting (CF) due to finetuning be avoided without changing training process, without any additional data, and without altering model structure. To this end the authors come up with Forgetting-Aware Pruning Metric (FAPM) wherein the task vectors are not only pruned based on magnitude but also uses the ratio of task vector to pretrained parameters to avoid CF. The authors provide extensive experiments across various tasks and latest LLMs like Qwen and Llama3 models.

**Strengths:**

1. The simplicity of the technique introduced by the authors is really great. Also I loved reading section 3.1 wherein they provide analysis and a clear thinking process of how they arrived at the method.

2. There are extensive experiments in the paper across various tasks and two models. The models chose are the latest ones which makes this paper a bit more relevant. Section 5 in general is pretty enjoyable to read.

3. Although the method is not pareto optimal but the numbers are consistently high on all the tasks across both Llama3 and Qwen2.

**Weaknesses:**

1. A small nitpick. It would be really great if the captions of the images and tables could be a bit longer and more informative.

**Questions:**

1. Nowadays most people prefer to use LoRA not because it reduces CF, but instead its pretty cheap. In my experience even LoRA suffers from CF if the finetuning domain is pretty niche. So I was wondering how informative are task vectors from LoRA merged models and how will this method perform if we use LoRA to finetune.

---

> ### Author Response · Authors · 2024-11-14
> **To Reviewer YFJm**
>
> Thank you for Reviewer YFJm's review.
>
> # Q1. A small nitpick. It would be really great if the captions of the images and tables could be a bit longer and more informative.
>
> We have revised the paper according to the reviewers' comments, with the following changes:
> 1. We have provided an explanation of the experimental setup for Figures 2 and 3.
>  2. We have improved the caption for Figure 4 by adding an example explanation.
>
> The revised paper has been re-uploaded.
>
> # Q2. Nowadays most people prefer to use LoRA not because it reduces CF, but instead its pretty cheap. In my experience even LoRA suffers from CF if the finetuning domain is pretty niche. So I was wondering how informative are task vectors from LoRA merged models and how will this method perform if we use LoRA to fine-tune.
>
> In our paper, we define task vectors as the subtraction of pre-trained weights from the weights fine-tuned on downstream tasks ($W_{ft} - W_{pre}$). In LoRA fine-tuning, the initialization of LoRA weights is not based on the pre-trained model weights. Therefore, the concept of task vectors as we defined cannot be applied when using LoRA for fine-tuning.
>
> However, through our consideration, we believe that with a simple modification, LoRA fine-tuning can still utilize our method. After training, we can merge the LoRA parameters with the pre-trained model parameters. The merging formula is $W_{new} = W_{pre} + W_{loraB}W_{loraA}$, where $W_{loraB}$ and $W_{loraA}$ are the LoRA matrices corresponding to a particular parameter matrix of the pre-trained model. Through the above formula, we observe that when merging the LoRA parameters with the pre-trained model parameters, it involves the dot product of the $W_{loraB}$ and $W_{loraA}$ matrices and then addition to the pre-trained model weights. Therefore, we can treat $W_{loraB}W_{loraA}$ as the task vectors in our algorithm for pruning. That is, $\Delta W$ in Equation 1 of the paper is $W_{loraB}W_{loraA}$.
>
> To validate the effectiveness of our approach, we conducted experiments on the SQuAD and MedQA datasets. MedQA is a newly added medical dataset. The model used is Llama3-8B. The experimental results are shown in the table below.
>
> | dataset | model    | ceval         | gsm8k         | mmlu          | humaneval     | Avg. | Performance |
> |---------------|---------------|---------------|---------------|---------------|------| ----- | ---- |
> | SQuAD| Pre-trained model | 0.4386        | 0.7922        | 0.6594        | 0.5914        | 0.6204 | 0.371|
> | SQuAD| LoRA          | 0.4795        | 0.7255        | 0.5914        | 0.5853        | 0.5954 | 0.648 |
> | SQuAD| FAPM (LoRA/0.9) | 0.4435        | 0.8013        | 0.6594        | 0.5853        | 0.6223 | 0.456 |
> | SQuAD| FAPM (LoRA/0.1) | 0.4502        | 0.7862        | 0.6552        | 0.5914        | 0.6207 | 0.616 |
> |SQuAD | FAPM (LoRA/0.05) | 0.4579        | 0.7793        | 0.6544        | 0.5886        | 0.6200 | 0.636 |
> | SQuAD| FAPM (LoRA/0.03) | 0.4591        | 0.7801        | 0.6511        | 0.5914        | 0.6204 | 0.644 |
> |||||||||
> | MedQA| Pre-trained model   | 0.4386        | 0.7922        | 0.6594        | 0.5914        | 0.6204 | 0.552 |
> | MedQA| LoRA                | 0.4216        | 0.7089        | 0.5889        | 0.5409        | 0.5650 | 0.640 |
> | MedQA| FAPM (LoRA/0.9)     | 0.4544        | 0.7899        | 0.6590        | 0.5914        | 0.6236 | 0.588 |
> | MedQA| FAPM (LoRA/0.1)     | 0.4540        | 0.7807        | 0.6439        | 0.5875        | 0.6165 | 0.623 |
> | MedQA| FAPM (LoRA/0.05)    | 0.4531        | 0.7845        | 0.6475        | 0.5914        | 0.6191 | 0.629 |
> | MedQA| FAPM (LoRA/0.03)    | 0.4592        | 0.7807        | 0.6484        | 0.5731        | 0.6154 | 0.632 |
>
> Table 1: Results of applying our method FAPM to LoRA fine-tuning on the SQuAD and MedQA dataset. The numbers after '/' indicate the pruning ratio. The 'Avg.' column represents the performance on general tasks, while 'Performance' indicates the performance on downstream tasks.
>
> From the table, we can see that LoRA fine-tuning exhibits some forgetting phenomena on both datasets, with the forgetting being more severe on the MedQA dataset. FAPM effectively alleviates forgetting to a mere 1% while maintaining an impressive 99% accuracy on downstream tasks in both datasets. Further experiments revealed that the pruning ratio in LoRA fine-tuning needs to be relatively small to avoid significant performance degradation in downstream tasks. We speculate that this is mainly because the LoRA matrices $W_{loraB}W_{loraA}$ are low-rank, and thus contain relatively few redundant parameters.

---

> > ### Comment · Reviewer_YFJm · 2024-11-22
> >
> > Thank you for the results. I would like to keep my rating as 6.

---

### Author Response · Authors · 2024-11-20
**General Response**

We thank the reviewers for their insightful comments and helpful feedback, which allowed us to significantly improve our submission. We are pleased that they highlighted the simplicity, effectiveness, and ease of implementation of our proposed method (YFJm, HmBx, aVtG) and the novelty of using the concept of pruning to address catastrophic forgetting in large language models (YFJm, HmBx, 53VB, aVtG). They mentioned that "Section 5 in general is pretty enjoyable to read and I loved reading section 3.1" (YFJm).

Here, we summarize our responses to the concerns raised by the reviewers, and we address the individual questions below each review.

***How our method can be applied to LoRA fine-tuning and its effectiveness:*** To validate the effectiveness of our method in LoRA fine-tuning, we conducted additional experiments. The results show that our method remains effective even when catastrophic forgetting occurs during LoRA fine-tuning. The experimental results are presented in Table 1 in our response to reviewer YFJm.

***Generalizability and instruction-following scenarios:*** We further validated the generalizability of our method in two highly specialized domains, namely medical and mathematical scenarios. The experimental results show that our method can maintain forgetting to just 1% while achieving 99% accuracy on downstream tasks in both scenarios. The detailed experimental results can be found in Table 1 and Table 2 of our response to reviewer HmBx, or in Table 3 and Table 4 of our response to reviewer aVtG.

***Ablation study:*** We conducted an additional ablation experiment using only the relative magnitude changes. The results demonstrate that using only relative magnitude changes leads to the inability to maintain progress on downstream tasks. The results can be found in Table 1 in our response to reviewer 53VB.

***Adding an Important Baseline:*** We have included an important baseline, "Wise-FT." Our experiments demonstrate that our method significantly outperforms Wise-FT in addressing the issue of catastrophic forgetting. The results can be found in Table 3 and Table 4 of our response to reviewer HmBx, or in Table 1 and Table 2 of our response to reviewer aVtG.

---

### Note · Authors · 2025-01-24

I have read and agree with the venue's withdrawal policy on behalf of myself and my co-authors.